# SnRNA sequencing defines signaling by RBC-derived extracellular vesicles in the murine heart

Nedyalka Valkov[1,*] , Avash Das[1,*] , Nathan R Tucker[1,5,6,*] , Guoping Li[1], Ane M Salvador[1], Mark D Chaffin[5], Getulio Pereira De Oliveira Junior[3] , Ivan Kur[2] , Priyanka Gokulnath[1] , Olivia Ziegler[1], Ashish Yeri[1], Shulin Lu[3], Aushee Khamesra[1], Chunyang Xiao[1], Rodosthenis Rodosthenous[1], Srimeenakshi Srinivasan[4], Vasilis Toxavidis[3], John Tigges[3], Louise C Laurent[4], Stefan Momma[2] , Robert Kitchen[1], Patrick Ellinor[1,5,†], Ionita Ghiran[3,†] , Saumya Das[1,†]

Extracellular vesicles (EVs) mediate intercellular signaling by transferring their cargo to recipient cells, but the functional consequences of signaling are not fully appreciated. RBC-derived EVs are abundant in circulation and have been implicated in regulating immune responses. Here, we use a transgenic mouse model for fluorescence-based mapping of RBC-EV recipient cells to assess the role of this intercellular signaling mechanism in heart disease. Using fluorescent-based mapping, we detected an increase in RBC-EV–targeted cardiomyocytes in a murine model of ischemic heart failure. Single cell nuclear RNA sequencing of the heart revealed a complex landscape of cardiac cells targeted by RBC-EVs, with enrichment of genes implicated in cell proliferation and stress signaling pathways compared with non-targeted cells. Correspondingly, cardiomyocytes targeted by RBC-EVs more frequently express cellular markers of DNA synthesis, suggesting the functional significance of EV-mediated signaling. In conclusion, our mouse model for mapping of EV-recipient cells reveals a complex cellular network of RBC-EV–mediated intercellular communication in ischemic heart failure and suggests a functional role for this mode of intercellular signaling.

## Introduction

Extracellular vesicles (EVs) are cell-derived membranous structures (100–1,000 nm in diameter) comprising exosomes and microvesicles (1). EVs carry diverse cargo including lipids, proteins, and RNA (2, 3) molecules that can be transferred to recipient cells (4) to mediate intercellular communication. Notably, miRNAs, generally known as

negative regulators of cellular mRNA expression (5) constitute a significant proportion of RNA present in EVs (6). Recent studies have shown that transfer of EV-miRNAs can subsequently alter target mRNA expression and the phenotype of recipient cells (7, 8).

Most of our understanding about EV function comes from studies using EVs derived either from cell culture conditioned media or biological fluids, and their subsequent administration in animal models to assess functional changes. This approach is less likely to reflect their in vivo composition and endogenous functions. Cells targeted by EVs and the functional consequences of delivery of EV cargo into those target cells in vivo remains largely unknown, mainly because of the lack of suitable tools and techniques to track EV targets. We have previously shown that functional *Cre* mRNA can be packaged in exosomes released by Cre recombinase–expressing cells and transferred to EV-recipient reporter cells, subsequently mediating *Cre*-dependent recombination to allow expression of a reporter (4). This system has allowed us to map targets of hematopoietic cell (HC) derived EVs at baseline and in a model of peripheral inflammation. More recently, our work has demonstrated that neuronal activity can influence the uptake of these EVs (9), suggesting an important role for the state of the recipient cell in this mode of signaling.

RBCs are a significant source of EVs in plasma. RBC-EVs play an immunomodulatory role with a demonstrated function in regulating monocyte-mediated activation of endothelial cells (10), modulating T-cell-monocyte immune synapse (11), and altering endothelial cells via transfer of functional microRNAs in the context of malarial infection (12). The release of RBC-EVs is triggered by membrane complement activation and calcium influx, providing a unique mechanism for the delivery of EVs at the site of inflammation (13). This mechanism of EV release can be leveraged to generate large quantities of pure RBC-EVs ex vivo for experimental use, making RBC-EVs attractive for mechanistic studies (13).

---

[1]Cardiovascular Research Center, Massachusetts General Hospital, Boston, MA, USA   [2]Institute of Neurology (Edinger Institute), University Hospital, Goethe University, Frankfurt am Main, Germany   [3]Beth Israel Deaconess Medical Center, Boston, MA, USA   [4]Department of Obstetrics, Gynecology, and Reproductive Sciences, University of California San Diego, La Jolla, CA, USA   [5]Cardiovascular Disease Initiative, Broad Institute, Cambridge, MA, USA   [6]Masonic Medical Research Institute, Utica, NY, USA

Correspondence: sdas@mgh.harvard.edu
*Nedyalka Valkov, Avash Das, and Nathan R Tucker contributed equally to this work
†Patrick Ellinor, Ionita Ghiran, and Saumya Das are co-senior authors

Adverse cardiac remodeling after myocardial infarction (MI) can eventually lead to heart failure (14), and comprises complex molecular, and cellular changes (15) mediated by multiple signaling pathways, and interactions between multiple cell types (16). Recent therapeutic strategies that target inflammation to reduce heart failure (17) suggest that interaction between HCs and cardiac cells may be mechanistically relevant in cardiac remodeling. Notably, recent evidence demonstrates the prognostic significance of RBC-EVs in post-MI patients (18, 19), inferring a direct role for miRNAs derived from circulating HCs in cardiac remodeling (20). However, the functional contribution of RBC-EVs as mediators of intercellular signaling (21) that contribute to post-MI LV remodeling remains unexplored, largely because of the limitations in tools for in vivo tracking of EVs (22).

Until recently, transcriptome studies in the heart to better understand the molecular underpinnings of cardiac remodeling have been performed using bulk heart tissue or biopsy samples containing a mixture of cell types. Although these studies have revealed important findings (23, 24), they do not reflect alterations in the proportions or gene expression profiles of individual cell types, or changes in gene expression in the most populous cell types, such as cardiomyocytes or fibroblasts. Importantly, pathology induced changes and cell type-specific signatures, particularly in less populous cell types, could be concealed. Single-cell RNA sequencing (RNA-seq) approaches have addressed this limitation by allowing for an in-depth characterization of complex tissues at the single-cell resolution and have provided important findings about cellular heterogeneity and identity (25, 26). However, conventional methods for single-cell RNA-seq on mammalian heart tissue are challenging because of the fact that cardiac tissues are difficult to dissociate without damaging constituent cells or inducing acute ischemia, thereby influencing cellular mRNA composition (27). In addition, the large size and irregular shape of the cardiomyocytes present challenge for microfluidics dependent approaches, resulting in their underrepresentation (27). As a complement to whole cell approaches, single cell nuclear RNA-seq (snRNA-seq), a more recently developed adaptation, has demonstrated ability to overcome these limitations to successfully profile cells from the heart (26, 28).

Here, we report the application of a fluorescence-based EV-target cell mapping technique based on the *Cre-LoxP* system (4) along with snRNA-seq to profile the role of RBC-EVs in a murine ischemic heart failure model. The EpoR-Cre transgenic mouse (*Cre* expression under the erythropoietin receptor promoter (29, 30)), when crossed with the Rosa26 mTomato/mGFP (31) mouse, leads to mGFP expression in RBCs, erythropoietic progenitor cells, and platelets to some degree (as they arise from megakaryocyte-erythrocyte precursors, MEPs) (32). In the absence of Cre expression, mTomato is expressed in all cells; only cells or tissues that express Cre in the double transgenic mice would have expression of mGFP. The RBCs in turn, generate mGFP$^+$ EVs that contain functional Cre protein. Transfer of functional Cre to target cells allows for identification of RBC-EV target cells in vivo. We leverage this EV-mapping model to study the targets of RBC-EVs at baseline and in a murine ischemic heart failure model (after ischemia/reperfusion/infarction or IR). Using snRNA-seq, we provide a detailed interrogation of cellular targets of RBC-EVs in the heart and assess differences in the transcriptome profiles between RBC-EV targeted and non-targeted cardiac cells in vivo. We show the

qualitative and quantitative alteration in RBC-EVs targets with IR and demonstrate the possible remote functional consequences of RBC-EV targeting. Taken together, our study is the first to show the distribution and target cell types of endogenous RBC-EVs in vivo and can be generalized for use by investigators to study the functional consequences of EV-mediated signaling.

# Results

## Murine model for fluorescence-based mapping of RBC-EVs target cells

To study EV-mediated communication between RBCs and different tissues, we crossed erythroid lineage-specific *Cre* knock-in mice (EpoR-Cre) (30) with membrane-targeted tandem dimer (td) Tomato/membrane-targeted GFP (mT/mG) mice (31) to generate double transgenic mice (EpoR-Cre/mTmG). In the absence of *Cre*, mTmG mice express Tomato protein on cell membranes of all cells. In cells where Cre recombinase is expressed (i.e., in the EpoR-Cre/mTmG mice), tdTomato (mTd) gene is excised, resulting in loss of tdTomato fluorescence and gain of mGFP expression (Fig 1A). As previously described (4), EVs from the *Cre*-expressing cells transfer functional Cre to EV-target cells, resulting in *Cre*-mediated recombination and expression of mGFP in the target cells. Furthermore, EVs from the *Cre*-expressing cells (in this case RBC-EVs) can be identified as GFP-positive EVs by fluorescence microscopy or nano-flow cytometry (33). Accordingly, fluorescence microscopy of the peripheral blood smear and cell-free plasma (CFP) from EpoR-Cre/mTmG double transgenic mice confirmed that all RBCs in the circulation and a substantial fraction of the EVs in plasma were mGFP$^+$, as expected (Fig 1B). Flow cytometry analysis of isolated blood cells (after removal of the buffy coat from peripheral blood) of EpoR-Cre/mTmG mice revealed the presence of Ter 119 (murine only, circulating erythroid-specific marker) in GFP$^+$ cells, confirming the presence of mGFP$^+$ RBCs in the peripheral circulation (Fig S1A). Ter 119 immuno-negative but GFP$^+$ cells were also seen, and likely represent contaminating platelets (see also Fig 1B) that originate from the MEPs (known to express EpoR) (32). Nano-flow cytometry of the plasma from EpoR-Cre/mTmG mice confirmed the presence of mGFP$^+$ EVs, whereas plasma from mTmG mice showed only mTd$^+$ EVs suggesting a large contribution of GFP$^+$-EVs from RBCs (Fig 1C) although some contribution from platelets cannot be excluded. Expectedly, no fluorescence was detected in the EpoR-Cre transgenic mice (Fig S1B). Quantification of EVs based on their fluorescence from EpoR-Cre/mTmG mice (mTd$^+$, mGFP$^+$, and mTd$^+$/mGFP$^+$) is represented in Fig S1C and showed significant a significant proportion of circulating EVs had mGFP positivity in the double transgenic mice. Double positive cells (expressing both mGFP and mTd) may arise either from recombination in one mTmG allele (but not the other) in homozygote mice, or from younger cells where the previously expressed mTd protein has not yet turned over.

We had previously demonstrated that EVs derived from *Cre*-expressing cells contain functional *Cre* mRNA that can be transferred to recipient cells (4, 9), mediating recombination and subsequent expression of a marker protein. To investigate whether Cre-recombinase is present in RBC-EVs, we first measured *Cre* mRNA and protein

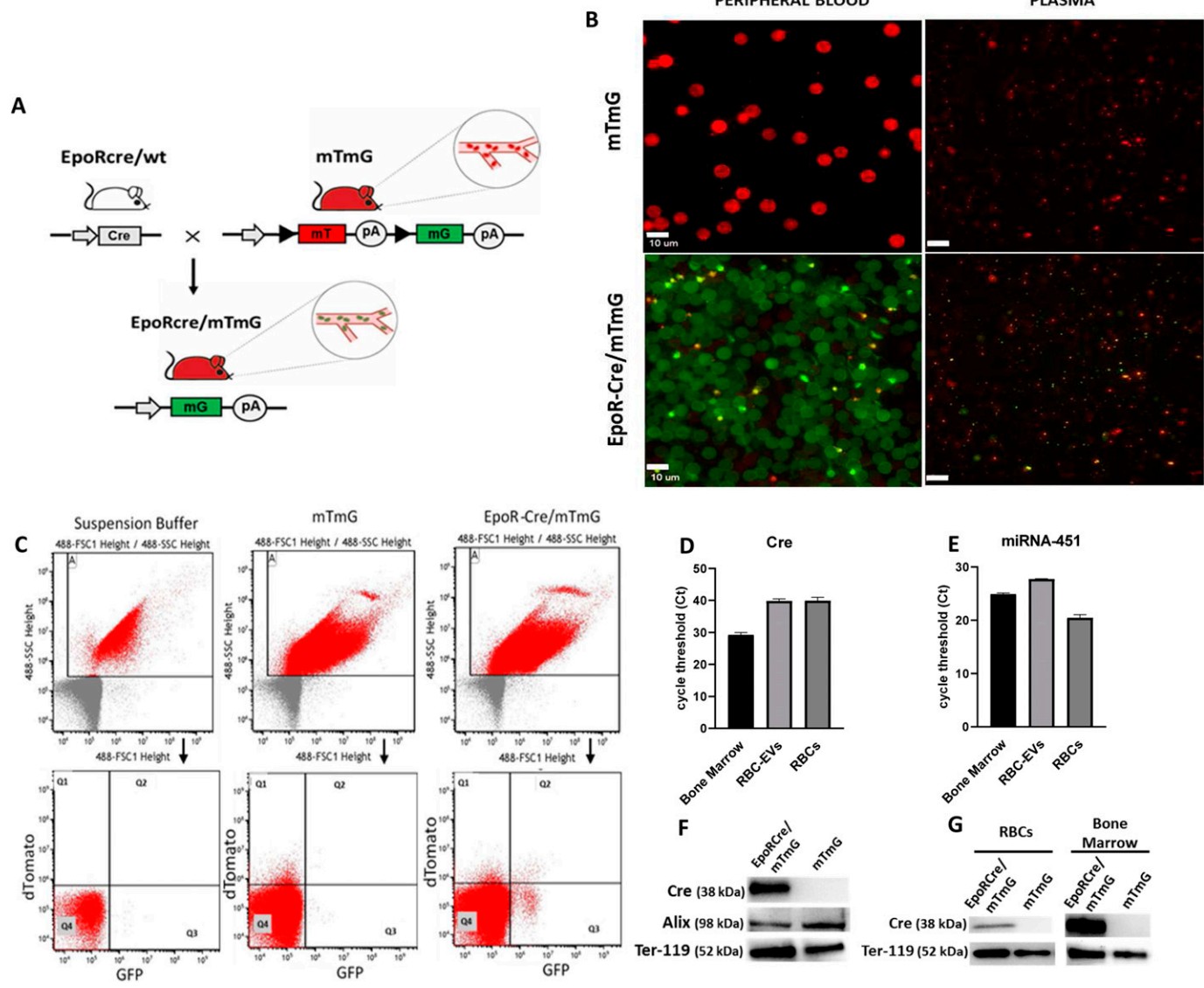

**Figure 1. Baseline characterization of RBC-extracellular vesicles (EVs) in transgenic murine model.**
**(A)** Schematic representation of the experimental murine models. Erythroid lineage specific conditional *Cre* knock-in mice (EpoR-Cre mice, BALB/c background) were crossed with membrane-targeted Tomato/membrane-targeted GFP (mTmG^Rosa26 mice, C57BL6/Sv129) to produce double transgenic (EpoR-Cre/mTmG) off-springs. Activated *Cre* mediates the removal of membrane-targeted tandem dimer Tomato (mT) sequence allowing the expression of membrane-targeted green fluorescent protein (mG) in RBCs. Arrows represent the direction of transcription. Triangles represent loxP sites for Cre-mediated recombination. pA stands for polyadenylation sequences. **(B)** High-magnification fluorescence images (63×) of fresh prepared peripheral blood smear and cell-free plasma smear from mTmG and EpoR-Cre/mTmG mice to characterize the mature circulating RBCs, and RBC-EVs (n = 4 in each group). Small bright GFP positive dots in the peripheral blood smear represent platelets. Scale bar: 10 μm. **(C)** Nanoflow cytometric characterization of cell-free plasma from mTmG mice and EpoR-Cre/ mTmG mouse plasma, showing presence of mGFP⁺ RBC-EVs in EpoR-Cre/ mTmG mouse plasma. The auto fluorescence from the Suspension Buffer (Phosphate Buffer Saline) has been accounted for, as background. Shown are Td on the y-axis and GFP in the x-axis and divided in four quadrants (Q1 is mTd+ [red]; Q2 is mTd+/mGFP+ [double positive]; Q3 is mGFP+ [green], and Q4 is background). **(D)** Real-time PCR analysis of Cre-recombinase mRNA in bone marrow, RBC-EVs and RBCs from EpoR-Cre mice. Data are shown as raw Ct values. n = 3 each. **(E)** Real-time PCR analysis of miRNA-451a in bone marrow, RBC-EVs and RBC from EpoR-Cre mice. Data are shown as raw Ct values. n = 3 each. **(F)** Western blots of RBC-EVs protein lysates from EpoR-Cre/mTmG and mTmG mice probed with the indicated antibodies: Cre-recombinase, Ter-119 (mature RBC- specific marker), and the exosome specific surface marker Alix. Data shown are representative of three replicated independent experiments. **(G)** Western blots of RBCs and bone marrow protein lysates probed with the indicated antibodies. Data shown are representative of three replicated independent experiments.

expression levels in RBC-EVs derived from EpoR-Cre/mTmG and mTmG mice. Real time PCR showed the presence of *Cre*-recombinase mRNA in bone marrow cells, but their absence in peripheral blood and RBC-EVs (Fig 1D and E), consistent with the general absence of mRNAs in RBCs (34). In contrast, miRNA-451, the abundant prototypical miRNA for RBCs,

was present in all three compartments (bone marrow cells, RBCs and RBC-EVs). To study the characteristics of RBC-EVs from wild-type or mice with cre expression in RBCs, we generated EVs ex vivo by exposing isolated RBCs from the mice to complement C5b-9 (Fig S2). RBC-EVs derived from the wild-type of cre-expressing mice were similar in size,

morphology and quantity as determined by nano-flow cytometry (Fig S2A), electron microscopy (Fig S2B) or tunable resistance pulse sensing measurements (Fig S2C). As previously noted, we confirmed presence of abundant miR-451 but lack of cre mRNA in these EVs by quantitative droplet digital PCR (ddPCR) (Fig S2D). Western blot confirmed the presence of Cre-recombinase protein in RBC-EVs, marked by the presence of Alix (Fig 1F), suggesting that Cre protein, present in RBCs and their precursors in the bone marrow (Fig 1G), is incorporated into RBC-EVs.

We next investigated whether *Cre*-recombinase protein contained within RBC-EVs can be: (i) transferred to recipient cells, and (ii) effectively mediate DNA recombination. Primary adult dermal fibroblast derived from mTmG mice (35) (all mTd$^+$ at the time of isolation), or a *Cre*-reporter HEK cell line were incubated with EVs isolated from EpoR-Cre or mTmG mouse CFP using density gradient ultracentrifugation. Our results show that after 48 h in the EpoR-Cre EV treatment group, both the dermal fibroblasts, as well as HEK cells underwent mTomato to GFP conversion, consistent with Cre-dependent recombination. No recombination events were recorded in the mTmG EV treated group, strongly indicating that the fluorescence conversion was in fact mediated by the transfer of functional *Cre* recombinase (Fig S1D).

### Mapping the target cell types of RBC-EVs

Recent studies suggest a role for EVs mediating intercellular communication (3, 36). To determine the cellular targets of RBC-EVs in our EV-mapping mouse model, we leveraged existing methodology for successful administration of EVs systemically in the Cre-reporter mice (4, 37). mGFP$^+$ RBC-EVs, generated ex vivo using complement activation of RBCs collected from EpoR-Cre/mTmG mice, were injected i.p. into wild type C57BL/6 mice. The presence of mGFP$^+$ EVs in the circulation was first recorded 48 h after i.p. administration demonstrating feasibility of EV infusions (Fig S3A). Next, complement-generated RBC-EVs from EpoR-Cre mice (shown to contain functional Cre protein, Fig 1F), or from mTmG control mice were transfused into the mTmG reporter mice. Tissue and plasma from the recipient mice were analyzed 7 d after i.p. administration (Fig 2A). Flow cytometry of the blood and nano-flow cytometry of cell-free plasma (Fig 2B and C) confirmed the presence of mGFP$^+$ RBCs in the circulation and mGFP$^+$ RBC-EVs in the plasma of the mTmG mice treated with EpoR-Cre RBC-EVs, but not with control mTmG RBC-EVs (which were not expected to contain Cre). EV quantification by nano-flow cytometry demonstrated that approximately 29.8% of EVs in the circulation of the mTmG mice transfused with the EpoR-Cre RBC-EVs were mGFP$^+$, and hence arose from mGFP$^+$ RBCs (Fig S3B). Together, these results demonstrated that i.p. delivery of RBC-EVs form EpoR-Cre mice results in recombination (likely in the nucleated RBC progenitor cells) and expression of mGFP in mature RBCs and EVs derived from them, thereby phenocopying the EpoR-*Cre*/mTmG double transgenic mice.

To determine the cell-types and organs targeted by RBC-EVs, we analyzed sections of other organs harvested from the EpoR-Cre/mTmG double transgenic mice. *Cre*-mediated recombination (mTd$^+$ to mGFP$^+$ transformation) was noted in cells in the heart, kidney, lungs, spleen and pericytes in brain, suggesting transfer of RBC-EV

molecular cargo (including Cre) at baseline (Figs 2D and S3C), whereas interestingly, the liver showed no recombination (Fig S3C). To exclude the possibility of "leaky" (38) or ectopic expression of *Cre* under the Erythropoietin (Epo) promoter leading to recombination in GFP$^+$ cells (39), or cell-fusion between circulating blood cells and the recombined cells (40) we also analyzed the relevant organs harvested from the mTmG mice transfused with EpoR-Cre RBC-EVs, as described above. These infusion experiments also allowed us to probe the targets of RBC-EVs specifically (as opposed to the double transgenic EpoR-Cre/mTmG mice that also had GFP$^+$ platelet-derived-EVs). The heart, kidney, lungs, and spleen showed the same profile of recombined cells as the EpoR-Cre/mTmG mice (Figs 2D and S3C), whereas the liver remained without any GFP-positive cells. A quantitative count profile of the mTd$^+$/mGFP$^-$(red) and mTd$^-$/mGFP$^+$(green) cells at baseline in EpoR-Cre/mTmG mice compared with EpoR-Cre RBC-EV transfusion in mTmG mice for these organs are shown in Fig S4 (40 high power field images for each organ per mice). These findings strongly suggest that recombination in these organs is mediated by the transfer of functional Cre cargo contained in RBC-EVs, and not via EV-independent mechanisms.

### RBC-EVs mediated signaling in post-infarct cardiac remodeling

RBCs have previously been implicated in vascular pathologies such as atherosclerosis (41), thrombosis (42), and inflammation (43). Moreover, EVs derived from RBCs increase under pathological stress (19). Based on our findings suggestive of EV-mediated cross-talk among RBCs and cardiomyocytes at baseline in unstressed mice, we hypothesized that RBC-EVs may play an enhanced role in intercellular signaling after the stress of MI, where increased generation of RBC-EVs because of complement activation at the site of coronary thrombosis or MI (44) would be expected. We assessed RBC-EV contribution to post-MI remodeling in EpoR-Cre/mTmG mice (45, 46) 4 wk after ischemia/reperfusion/infarction (IR, Fig 3A).

Mice in the IR group but not the sham control group demonstrated ventricular dysfunction as measured by a decreased fractional shortening (FS) by echocardiography (Fig 3B). Nano-flow cytometry of plasma did not show any significant difference in the count profile of plasma total EVs and RBC-EVs (mGFP$^+$ EVs) between the IR and sham groups mice at the 4-wk time point (Fig S5A–C). Cardiomyocytes isolated from the hearts of these animals and subjected to flow cytometry showed a significant increase in the number of recombined (mGFP$^+$) cardiomyocytes in the IR group compared with their sham counterparts or wild-type unoperated mice (Figs 3C and D and S5D). At the 4-wk stage in the remodeling process, there was no distinct spatial grouping of the recombined GFP$^+$ cardiomyocytes, with no significant differences noted between infarct, peri-infarct area or remote areas, suggesting that when assessed at a time-point corresponding to chronic post-MI remodeling, RBC-EVs had targeted cardiomyocytes both in the peri-infarct and remote zones (Fig 3E). Microscopic count profiles of the kidney, lung, and spleen did not show any significant difference among IR, sham, and baseline groups (Fig S5E).

We have previously demonstrated communication between HCs and brain cells (including neurons and microglia) in the setting of

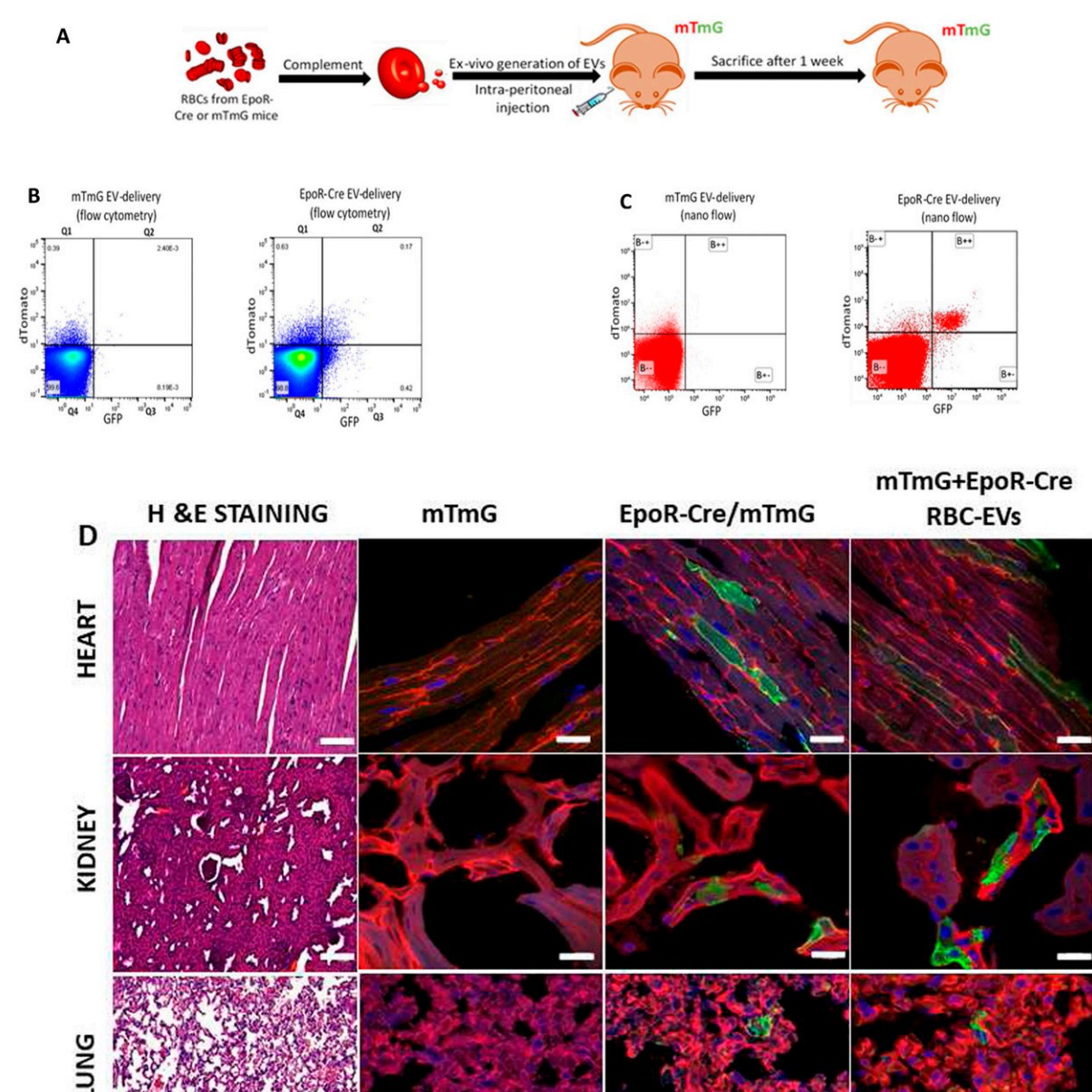

**Figure 2. In vivo transfer of functional Cre-recombinase through RBC-extracellular vesicles (EVs).**

**(A)** Schematic representation of experimental design for i.p. injection in mTmG mice with complement-generated RBC-EVs. **(B)** Flow cytometry scatterplots of peripheral blood from mTmG mice transfused with mTmG RBC-EVs and EpoR-Cre RBC-EVs. Shown are Td on the y-axis and GFP in the x-axis and divided in four quadrants (Q4 is background, Q1 is mTd [red], Q3 is mGFP+ [green], and Q2 is mTd+/mGFP+ [double-positive]). **(C)** Nanoflow cytometric characterization of cell-free plasma from mTmG mice transfused with EpoR-Cre RBC-EVs. Shown are Td on the y-axis and GFP in the x-axis and divided in four quadrants (B− is background, B-+ is mTd+[red], B+- is mGFP+ [green], and B++ is mTd+/mGFP+ [double-positive]). **(D)** Low-magnification fluorescence microscopy (20×) of paraffin embedded hematoxylin and eosin–stained sections

peripheral inflammation and physiological neuronal activity (4, 9). Given recent attention on the association between ischemic heart disease and neuro-cognitive disorders (47), we next investigated the putative intercellular communication between RBC-EVs and brain cells during ischemic heart disease. Our results show that 4 wk after IR surgery there was *Cre*-mediated recombination in microglia cells of the cerebellum and hippocampus of EpoR-Cre/mTmG mice, but not sham surgery (Fig 3F and G). The brain cells targeted by RBC-EVs appear to differ from the targets of other HCs: recombination is seen in neurons in addition to microglia in the vav-iCre/mTmG mouse model where Cre is expressed in vav-expressing cells that include macrophages and monocytes after peripheral inflammation (4) or IR (Fig S5F). Together, these data suggest that during inflammatory conditions, the cellular origin of EVs may in part determine their target cells.

To confirm whether the observed pattern of communication between RBCs and cardiomyocytes after IR is mediated by RBC-EVs and not a result of cell-fusion or ectopic expression of *Cre* after MI (40, 48), we performed IR on recipient mTmG mice injected with *Cre*-containing RBC-EVs (harvested from EpoR-Cre RBCs) during reperfusion (Fig S6A). Similar to our previous results, 4 wk after MI we observed mGFP$^+$ cells that appeared to be cardiomyocytes in the IR model strongly suggesting that transfer of functional cre recombinase from RBC-EVs rather than cell fusion, or MI-induced activation of cre leads to the observed recombination events in cardiomyocytes (Fig S6B and C).

### Single nucleus RNA-sequencing of the adult EpoR-Cre/mTmG mouse heart

To define the cellular targets of RBC-EVs in the heart after MI and further investigate the transcriptional changes induced in recipient cells by the RBC-EVs, we performed single nuclear RNA sequencing (snRNA-seq). Ventricular tissue samples taken from EpoR-Cre/mTmG mice undergoing IR or sham surgery were subjected to nuclear isolation and processing for snRNA-seq. From the ventricles of six mice we captured and sequenced a total of 42,497 nuclei, with a median of 1,064 genes and 1,930 unique transcripts per nucleus. Clustering analysis of the resulting transcript matrix revealed 13 distinct cell types visualized as a UMAP representation (Fig 4A). The largest clusters can be attributed to populations of fibroblasts (10,905 nuclei, 25.7%), cardiomyocytes (9,865 nuclei, 23.2%), and endothelial cells (8,954 nuclei, 21.1%). Based on the expression pattern of canonical cell-type-specific marker genes in the cluster, each cluster was assigned a cell type label (Fig 4B and Table S1). Importantly, in addition to observing the major cell types of the heart, we also identified and transcriptionally profiled various immune subtypes, cells of neuronal origin (0.4%), and adipocytes (0.07%).

We next sought to determine cardiac cell types targeted by RBC-EVs in the EpoR-Cre/mTmG mice by assessing the presence of

mTomato (non-recombined) or mGFP (recombined) transcripts. *Cre*-mediated recombination (mTd+ to mGFP$^+$ switch) was noted in all cell clusters identified in the heart (although at different recombination frequencies), indicating transfer of RBC-EV cargo (including Cre) (Fig 4C and Table S2). We then compared the proportion of mGFP$^+$ with the mTd$^+$ by cell type in all the animals including only cells that were exclusively mGFP$^+$ (2,391 total) or mTd+ (3,206 total) in the calculation. As shown in Table S2, the proportion of mGFP+ cells in animals subjected to MI vary among cell types, suggesting that the various cell types of the heart differ in their ability to uptake RBC-EVs. For example, based on proportion of mGFP$^+$ to mTd$^+$ by cell type, a large number of macrophages (62.8%), B-cells (63.8%) and lymphatic endothelial cells (68.9%) were targeted by RBC-EVs, whereas only few vascular smooth muscle cells (18.2%) demonstrated recombination. Cardiomyocytes are intermediate (49.5%) in terms of being targeted by RBC-EVs.

We next assessed if the degree of ventricular dysfunction affects RBC-EV–mediated intercellular signaling to cardiomyocytes specifically. Echocardiographic measurements (Fig 4D) showed that IR induced ventricular dysfunction in the EpoR-Cre/mTmG mice as indicated by decrease in FS in the IR group compared to sham 4 wk after IR. There was a strong negative correlation between FS and mGFP$^+$ cardiomyocytes, suggesting that animals with a greater degree of myocardial dysfunction after MI have a greater number of recombined cardiomyocytes (Fig 4E), consistent with the observation from our flow cytometry analysis above (Fig 3C and D). Together, these data suggest that intercellular signaling mediated by RBC-EVs increase notably upon cardiac stress.

We next sought to determine whether the transfer of RBC-EVs cargo is associated with changes in the transcriptome of the recipient cells. We performed differential expression testing on all the cell types and found several genes that were differentially expressed between mGFP$^+$ and mTd$^+$ cells, including genes implicated in regulation of cardiac function (*Mhrt* and *Sema3a*) (49,50), cell-cycle regulation and cardiac hypertrophy (*Ccnd3*) (51), and regulation of splicing in cardiomyocyte-specific genes (*Rbm20*) (52). Given the small number of significant differentially expressed genes from the global analysis in this pilot study, we chose the 743 genes with an unadjusted *P*-value ≤ 0.05 between the two groups to perform network analysis. Of these (Table S3), we identified genes that are involved in regulation of cardiac function (*Mhrt*, *Sema3a*, *Rbm20*, and *Obscn*) cell cycle and proliferation (*Arhgap24*, *Prkg1*, *Vav3*, and *Ccnd3*) and DNA repair and metabolism (*Lrp1*, *Phf21a*, *Nox4*, and *Dusp27*). A network analysis to determine pathways that are differentially regulated in cells that received the cargo of RBC-EVs identified significantly enriched pathways involved in cell cycle, dilated cardiomyopathy and adrenergic signaling (Fig 4F). Whereas these data serve as a pilot to demonstrate the feasibility of using snRNA-seq on the exosome-tracking mouse model, our findings raise the possibility that the transfer of RBC-EV cargo can confer

of the heart, kidney, lung, and spleen from mTmG, EpoR-Cre/mTmG, and mTmG mice transfused with EpoR-Cre RBC-EVs to characterize organ-specific tissue architecture. Corresponding, high-magnification (63×) confocal microscopy of fixed frozen section of heart, kidney, lung and spleen from mTmG, EpoR-Cre/mTmG, and mTmG mice transfused with EpoR-Cre RBC-EVs to map their fluorescence profile (mTomato [red] and mGFP [green]) at baseline and after transfusion. The findings were separately verified using biological replicates (n = 4 for mTmG, n = 6 for EpoR-Cre/mTmG and n = 4 for mTmG mice transfused with EpoR-Cre RBC-EVs). Scale bar: 10 μm.

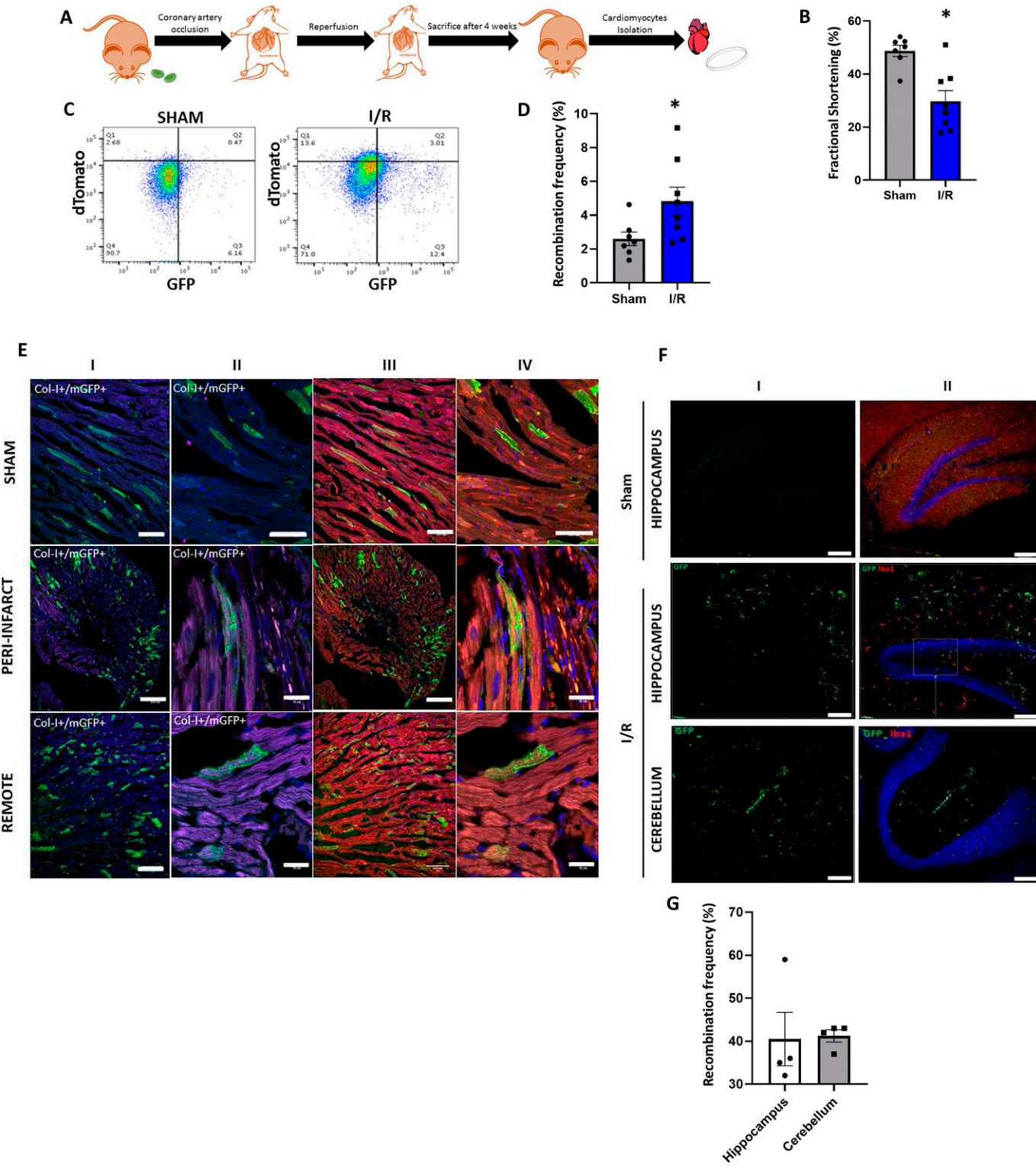

**Figure 3. RBC-extracellular vesicle–mediated intercellular communication in ischemic heart failure.**
**(A)** Schematic representation of experimental design for EpoR-Cre/mTmG mice undergoing IR. **(B)** Box plots quantifying fractional shortening in sham and IR groups of EpoR-Cre/mTmG mice (29.7% ± 1.97% versus 48.73% ± 1.97%, paired *t* test, *P* = 0.015), normality established by Shapiro–Wilk (n = 8 for IR and n = 7 for sham). Data are presented as mean ± SEM. **(C)** Representative flow cytometry scatterplots of CMs (viable, DAPI) from sham and IR EpoR-Cre/mTmG mice comparing the mGFP+ CMs between both groups. Shown are Td on the y-axis and GFP in the x-axis and divided in four quadrants (Q4 is background, Q1 is mTd+ [red], Q3 is mGFP+ [green] and Q2 is mTd+/mGFP+[double-positive]). (n = 7 for sham and n = 8 for IR). **(D)** Box plots quantifying the recombination frequency of CMs in the sham and IR groups of EpoR-Cre/mTmG mice (2.60% ± 0.37% versus 4.82% ± 0.79%, paired *t* test, *P* = 0.013, normality tested by Shapiro–Wilk, n = 8 for IR and n = 7 for sham). Data are presented as Mean ± SEM.

functional changes in the transcriptional profile of the recipient cells that may affect cardiac remodeling.

## Transcriptional changes mediated by RBC-EV–mediated signaling in vitro

To directly study the transfer of RBC-EV cargo on changes in the transcriptome of recipient cells, we studied the effect of RBC-EVs on a mT/mG HEK293 reporter cell line. EVs purified from the plasma of either EpoR-Cre or mTmG mice by differential gradient ultracentrifugation (Fig S7A and B) were directly applied to the mTmG reporter HEK293 cell line. We observed Cre-mediated recombination (appearance of mGFP$^+$ cells) only in the EpoR-Cre EV treatment group and not in mTmG EV-treated group (Fig S1D, bottom panel). Using flow sorting, we separated mTd$^+$, mGFP$^+$, and mTd$^+$/mGFP$^+$ sub-populations from mT/mG HEK293 treated with EpoR-Cre and mTmG EVs (Fig S7C). Overall, recombination efficiency in this model was low: mGFP$^+$ and mTd$^+$/mGFP$^+$ cells constituted 5.72% ± 1.51% of the total population of flow sorted mT/mG HEK293 treated with EpoR-Cre EVs (Fig S7D). Targeted whole transcriptomic sequencing (Ion gene Studio S5 Next Generation Sequencing Systems) identified 143 genes that were expressed with at least 25 reads per million (RPM) at a twofold change between the two groups (mTd$^+$ as one group versus mGFP$^+$ and double positive mTd$^+$/mGFP$^+$ grouped together). The regulation of different molecular pathways by these 143 genes is shown by Ingenuity Pathway Analysis (Fig S7E). We identified differentially expressed genes from relevant biological pathways such as cyclin and cell cycle regulation, cell cycle checkpoint control, DNA repair and phosphorylation. Although these cell culture experiments were on a different cell type, some of these pathways were also identified in the snRNA-seq analysis of recombined and non-recombined heart cells from the EpoR-Cre/mTmG mouse model. Taken together, our results suggest that the cargo of RBC-EVs can mediate transcriptional changes that may affect important cellular processes in recipient cells.

## Functional effects of RBC-EVs in ischemic heart failure

Cardiomyocytes have been shown to proliferate at low frequencies (53), and can re-enter the cell cycle after MI (54). Because the change in the transcriptome from the snRNA-seq on the heart cells and HEK293 cells receiving RBC-EVs suggested a potential role of RBC-EV–mediated signaling in regulation of cell cycle pathways, we explored the possibility of a similar role in the heart during post-MI cardiac remodeling. IR was performed on EdU injected EpoR-Cre/mTmG mice and euthanized after 4 wk when they had developed ventricular dysfunction (Fig S8A). In the heart, EdU$^+$ cells were

detected in higher numbers near the infarct zone compared with the remote zone in IR mice (Fig S8B). Although the overall frequency of EdU+ cardiomyocytes was low, consistent with other studies, we found that EdU$^+$ cardiomyocytes were represented at a significantly higher (twofold) proportion in GFP$^+$ recombined cardiomyocytes compared with mTd$^+$ cardiomyocytes (Fig 5A). Complementary to these results, immunostaining of heart sections from IR EpoR-Cre/mTmG mice with another proliferation marker, Phospho-histone H3 (PHH3), showed a similar pattern of PHH3-immunoreactivity in mGFP$^+$ cardiomyocytes (Fig 5B and C). Together, our data suggest that transfer of RBC-EV cargo constituted a novel signaling paradigm in cardiac remodeling that could alter the transcriptional profiles of target cells and enhance DNA synthesis.

## RBC-EV RNA cargo may mediate intercellular signaling

Previous studies have demonstrated the importance of EV cargo RNAs, notably miRNAs in modulating recipient cell function (3). We therefore sought to determine the small RNA content of RBC-EVs and their possible regulation of pathways highlighted by the snRNA-seq described above. Human RBC-EVs can be isolated using immunoaffinity-bead based flow sorting with a validated antibody against Glycophorin-A; the lack of a reliable antibody against the equivalent antigen on mouse RBC-EVs precluded these experiments in our murine model. We explored RBC-EV content in a clinically well-characterized biorepository of human plasma from patients with MI and control patients referred for supraventricular tachycardia (SVT). After confirming the specificity of the Glycophorin-A positive EV isolation method using flow cytometry (Fig S9A), microscopy (Fig S9B) and digital PCR for RBC-enriched RNAs (Fig S9C), we performed small RNA-seq to analyze RNA content in Glycophorin-A (CD235a)-immunopositive EVs from the plasma of human subjects (see clinical characterization of the patients in Table S4).

After analyzing pooled RNA-seq data for samples that passed minimal QC (see the Materials and Methods section), we identified a total of 105 distinct miRNAs in the plasma derived RBC-EVs (see Table S5 for counts of miRNAs across all patients). Among them, however, only a few miRNAs were highly expressed. miRNA-451a was the most enriched, comprising 11.7% of the total miRNA signal in terms of RPM mapped to the transcriptome, followed by miR486-5p (9.4%) and let-7a-5p (2.70%) (Fig S10A). There were no significant differences in the small RNA content of RBC-EVs from the patient cohorts (MI or SVT) as shown in the heat map (Fig S10B). The targets of the top 20 miRNAs were identified using TargetScan v 7.1 and all target mRNAs with a cumulative context score less than 0.2 were included for Gene Ontology analysis using DAVID (55). We identified enriched pathways involved in cell cycle, cell division, cell migration

---

**(E)** Representative fluorescent images of frozen ventricular heart sections of EpoR-Cre/mTmG IR mice from sham, remote and peri-infarct areas. (I) Low-magnification (20×) and (II) corresponding high-magnification (63×) images of heart sections stained with Collagen I (stains for fibrosis, pseudo-color Magenta) and DAPI staining of cell nucleus (blue). (III) Low-magnification (20×) and (IV) corresponding high-magnification (63×) images of heart sections stained for mTomato (red) and mGFP (green). Scale bar: 0.01 and 10 cm for low- and high-magnification images, respectively. Scale bar: 0.01 cm for 20× and 10 cm for 63×. Scale bar: 0.01 and 10 cm. **(F)** High-magnification (63×) confocal microscopy along with 3D reconstruction and 2D projection of images from fixed paraffin-embedded brain tissue sections form EpoR-Cre/mTmG mice undergoing IR. (I) Mapping of nuclei (DAPI, stained blue) and GFP+ cells (green) in hippocampus and cerebellum of EpoR-Cre/mTmG mice undergoing IR (II) Immunostaining of the sections with Iba1 antibody (pseudo-color red, co-staining for resident microglia) and corresponding overlay of PE (red) channel to demonstrate Iba1+/mGFP+ cells (n = 4). Scale bar: 10 $\mu$m. **(G)** Box plot quantifying fraction of Iba1+/mGFP+ microglia compared to total Iba1+ cells in the hippocampus (41% ± 5%) and cerebellum (41% ± 0.1%) of EpoR-Cre/mTmG mice after IR (n = 4). Sham mice had no GFP+ microglia. Data are presented as mean ± SEM.

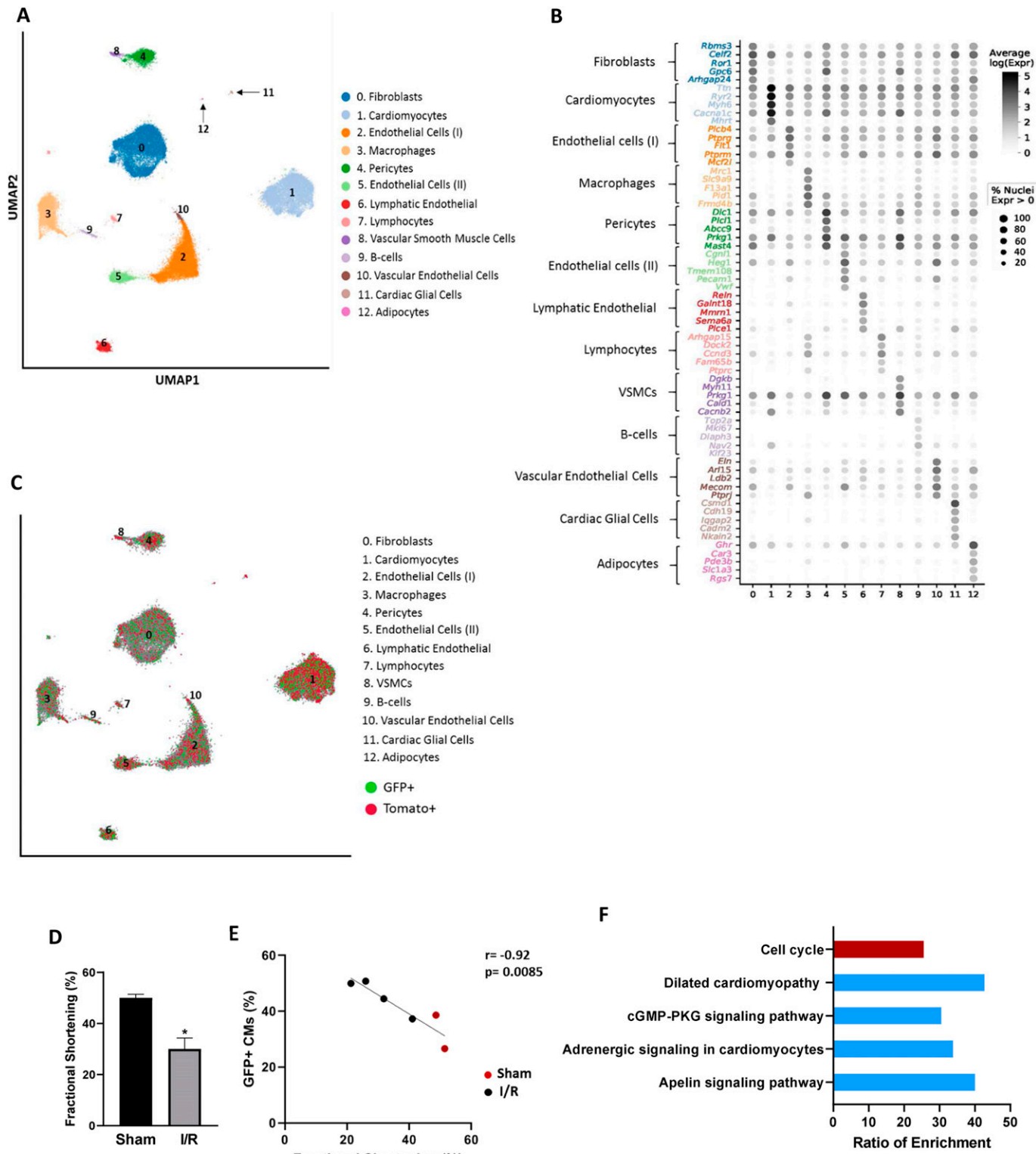

**Figure 4. Single-nucleus RNA-sequencing analysis in the adult EpoR-Cre/mTmG mouse heart.**
**(A)** UMAP clustering reveals several distinct clusters in the heart of the EpoR-Cre/mTmG mouse undergoing I/R for the indicated cell types. Clusters are colored by inferred cell type and each dot represents an individual cell. **(B)** Dot plot of gene expression marker genes for the identified cell types. The size of the dots denotes the relative gene expression in percent for each cluster, for example, 100% means every cell within this cluster express this gene. The color shows the average expression level for the indicated gene per cell type. The numbers of the clusters on the x-axes are taken from Fig 4A. The brackets represent the top 5 marker genes for the corresponding cell cluster. **(C)** UMAP overlaid with GFP and Tomato for each cell cluster. **(D)** Bar graph quantifying fractional shortening in the sham and I/R groups of

and proliferation (Fig S10C). Moreover, some of the genes, found to be differentially expressed between mGFP+ and mTd+ cells in the snRNA-seq analysis, were identified as potential targets of miRNAs detected in the RBC-EV in the patient plasma (Fig S10D). Notably, Sema3a is a potential target of miR-30e and miRNA-21, whereas the latter also targets Arhgap24, another top differentially expressed gene in the snRNA-seq. These data suggest that whereas RNA content of RBC-EVs appears to be consistent across patients, generation of RBC-EVs, access to cardiac cells, and uptake of these EVs into recipient cells may regulate genes and important biological pathways in recipient cells.

## Discussion

EVs have recently been shown to constitute a novel mode of intercellular signaling by transferring their cargo molecules to recipient cells. RBCs are the most abundant cell type in the circulation, and RBC-EVs comprise a significant proportion of circulating EVs in human plasma (11). RBC-EVs have recently been recognized as key regulators of various physiological and pathological processes, including coagulation (11), inflammation, atherosclerosis, and thrombosis (56, 57). In patients with heart failure, MI or infections such as malaria, the levels of RBC-EVs are often found to be increased (19). Importantly, complement activation at the site of inflammation makes RBC-EVs (13) poised to fuse with, and deliver their cargo to recipient cells. However, despite increased appreciation for their biological importance, mechanistic insight into the functional consequence of the RBC-EV uptake into recipient cells is largely unknown.

In this study, we adapted a murine model previously used to characterize the targets of HC-derived EVs, to specifically study the role of RBC-EVs in ischemic heart failure. Using a double-fluorescent *Cre* reporter that expresses membrane targeted tandem dimer tomato (mT) globally, but expresses membrane-targeted EGFP (mGFP) after Cre-mediated recombination (4), we mapped the cellular targets of RBC-EVs that were identified by the expression of mGFP after transfer of EV-mediated functional Cre protein using several orthogonal technologies, including snRNAseq.

*Cre* expression is restricted to erythroid progenitor cells in the EpoR-Cre mice, leading to GFP positivity in all progeny including reticulocytes and mature RBCs. Platelets, which arise from the common MEP cells that also express EpoR, show evidence of mGFP positivity; however, given that megakaryocytes do not express EpoR, although erythrocyte progenitors continue to express EpoR, the efficiency of recombination is likely much higher in the RBC lineage as seen in our flow cytometry and microscopy data. Importantly, we found that Cre protein (and not mRNA) was packaged into RBC-EVs and was functional upon transfer to recipient cells not only in cell culture models but also in vivo. The fact that RBCs differ from other cells in not having a nucleus may account for the fact that Cre protein is packaged from RBC cytoplasm into the EVs. Identification of the RBC-EV target cells by the Cre-mediated fluorescence

conversion of the target cells allowed us to study EV-mediated intercellular communication between RBCs and different cell-types both at baseline and in a murine ischemic HF model. Importantly, our EV transfusion experiments confirmed that recombination in cells other than erythroid cells was due to transfer of RBC-EV cargo Cre, rather than transfer from platelet-derived EVs, ectopic Cre expression or cell fusion (4). Nonetheless, it is theoretically possible that RBC-remnants or non-EV cre mRNA or protein may be taken up by cell-types that show recombination.

Multiple different cell types were targeted by RBC-EVs, and included cardiomyocytes, tubular epithelial cells in the kidney and splenic cells, but interestingly, not cells in the liver, at least as measured by transfer of functional cre protein. In the case of liver, it is possible that RBC-EVs were internalized by the reticuloendo-thelial system (RE) system (Kupffer cells), and destroyed by the lysosomes, thus preventing the Cre to access the nucleus and mediate recombination. Whether surface markers on RBC-EVs, or specific EV "receptors" on these cells are responsible for the observed pattern is not clear at this point. In addition to the observed recombined and non-recombined cells, the microscopy data also revealed the presence of some double-positive cells ("yellow"). This observation could be explained by the persistence of the tdTomato transcript in cells that had recently recombined, leading to double positive cells even after recombination. These microscopic observations are supported by the snRNA-seq data that notes the presence of a small number of "double-positive" cells. In the case of cardiomyocytes, double-positive cells could be also observed in binucleate cells as one nucleus may have had recombination, whereas the other has not. Overall, baseline levels of recombination are low, suggesting a modest to low level of signaling mediated by RBC-EVs in the absence of stress. Whether the uptake at baseline is a stochastic process or reflects a distinct functional hierarchy in target cells remains to be determined.

To determine in more detail the in vivo targets and functions of RBC-EVs in the heart specifically, we used snRNA-seq to characterize the targets of RBC-EVs within the heart and started delineating the transcriptional changes between RBC-EV targeted and non-targeted cardiac cells. Compared with conventional FACS-based methods and single-cell RNA-seq, snRNA-seq offers several advantages when applied to technically challenging organs and tissues like the heart or brain. SnRNA-seq overcomes the hurdle of extracting intact cells from tissues that are difficult to dissociate, minimizes biased recovery of easily dissociated cell or cells of certain size, and reduces aberrant transcriptional changes during cell extraction (58). In this study, we show feasibility of using this tool to successfully map the RBC-EV cell targets in the heart and begin elucidating the transcriptional profile of the recipient cells. We identified a total of 13 distinct cell clusters, with the largest clusters attributed to cardiomyocytes, fibroblasts, and endothelial cells, which correlates with other recent findings, also based on single-nucleus sequencing (28); the cell-type profiles were also

EpoR-Cre/mTmG mice n = 4 for I/R and n = 2 for sham. **(E)** Correlation of the percentage of recombined CMs (GFP+) and the fractional shortening in the sham and I/R group of EpoR-Cre/mTmG mice n = 4 for I/R and n = 2 for sham. **(F)** Bar chart of enrichment ratios for KEGG category. Ratio of enrichment = the number of observed genes divided by the number of expected genes from KEGG. Red color represents up-regulated pathways and blue color represents down-regulated pathways. KEGG, Kyoto Encyclopedia of Genes and Genomes.

**A**

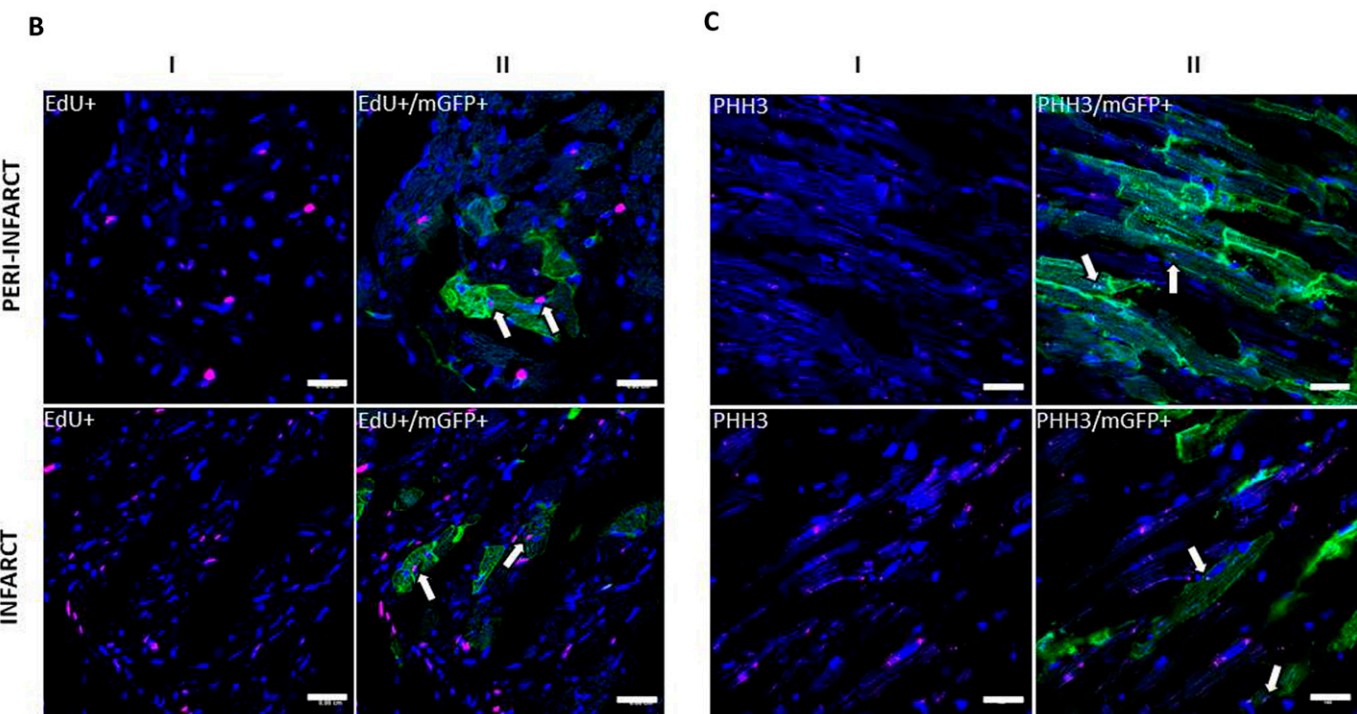

**Figure 5. RBC-extracellular vesicles modulate post-infarct cardiac remodeling.**
**(A)** Box plot comparing EdU+/mTd+ and EdU+/mGFP+ CMs in the EpoR-Cre/mTmG mice undergoing I/R (11.19% ± 2.18% versus 18.89% ± 2.82%, paired *t* test, *P* = 0.056, Shapiro–Wilk test passed for normality). n = 3. Data are presented as Mean ± SEM. **(B)** Representative fluorescent images of frozen ventricular heart sections of Edu injected EpoR-Cre/mTmG mice with I/R from infarct and peri-infarct areas. (I) High-magnification (63×) images showing EdU–positive cell nuclei (Alexa 647, pseudo-color Magenta) and nuclei stained with DAPI (blue) highlighting their overlap. (II) corresponding overlay of GFP channel demonstrating EdU+/mGFP+ CMs. n = 4 for I/R and n = 3 for sham. Scale bar: 0.01 cm. **(C)** Representative fluorescent images of frozen ventricular heart sections of EpoR-Cre/mTmG mice with I/R from infarct and peri-infarct areas. (I) High-magnification (63×) images showing PHH3 positive cells (Alexa 647, pseudo-color Magenta) and nuclei stained with DAPI (blue) highlighting their overlap and (II) corresponding overlay of GFP channel showing PHH3+/mGFP+ CMs. Scale bar: 0.01 cm.

consistent with published cardiac cell markers. Importantly, we identified several relatively rare cell types, including various immune cells, pericytes, glial cells, smooth muscle cells, and adipocytes, representing the cellular heterogeneity and complexity of the heart and indicating the robustness of our data.

We found that different cardiac cells were targeted by RBC-EVs, including cardiomyocytes, fibroblasts, and endothelial cells, but that the proportion of recombined cells (mGFP+ cells) varied greatly among cell types, suggesting differential uptake of the RBC-EVs by the various cells in the ventricles of the IR animals. The finding that cardiac resident cells other than cardiomyocytes are targeted by RBC-EVs is intriguing. What determines the observed preference and different recipient cells' RBC-EVs uptake capabilities is currently unclear. It could be that RBC-EVs display proteins and other

signals that may confer selective uptake and/or RBC-EVs uptake could be affected by changes in the status of the cells upon alterations in cell environment (e.g., during cardiac stress). The observation that a large number of immune cells like macrophages, lymphocytes, lymphatic endothelial cells, and B-cells received RBC-EVs points to the role of immune regulation of the post-ischemic remodeling process (59, 60). Immunomodulatory effects and interactions of RBC-EVs with various cells of the immune system have been reported in a number of studies. For instance, RBC-EVs have been shown to induce a respiratory burst and increase the ability of neutrophils to phagocytose (61), interact with leukocyte and platelets to increase inflammatory chemokine bioavailability (62), and to activate endothelial cells in the presence of monocytes (10). Whether transfer of RBC-EVs to these cells, especially during MI when immune cells are activated and RBC-EV levels are elevated at the site of complement activation (44), causes alterations in their signaling pathways and phenotypes is an interesting avenue for research that warrants further investigation. Previous studies had suggested that EV-mediated signaling is implicated in disease pathogenesis, such as cancer cell metastasis (63), immune escape of tumor cells (64), and obesity-associated insulin resistance (65).

We found an increase in the number of recombined CMs in the IR model suggesting that similar to the model previously described from the vav-iCre system, RBC-EV-mediated signaling can increase markedly after the stress of MI. Importantly, the number of recombined cardiomyocytes was directly and significantly correlated with cardiac dysfunction, suggesting that RBC-EV–mediated signaling could increase upon cardiac stress. The timing of these signaling events were not elucidated as we only examined the 4-wk time point, at which time we found recombined cardiomyocytes not only at the peri-infarct zone (as expected) but also in remote myocardium. Similar observations have been made when using Cre-recombinase–based tracing of EVs in the tumor microenvironment (36, 66), indicating the importance of the state of the recipient cell, rather than the sole presence of EVs. Finally, we found that recombined cardiomyocytes were more likely to undergo DNA synthesis than non-recombined cardiomyocytes. Whether the increased DNA synthesis noted in cardiomyocyte recipients of RBC-EVs primes them to re-enter the cell cycle or allows for DNA synthesis and repair after oxidative damage from reperfusion would be the topic of future investigations.

The RNA cargo of EVs has garnered significant attention, as active signaling moieties. Our deep RNA sequencing of RNA isolated from RBC-EVs from human subjects revealed that the content of RBC-EVs is dominated by a small number of miRNAs, and that this content does not change appreciably in patients with MI. We found that the pathways predicted to be regulated by these miRNAs are implicated in cell cycle regulation, proliferation and oxidative metabolism. These same pathways were implicated in the genes differentially expressed between cells targeted by RBC-EVs and non-targeted cells in both in vitro and in vivo experimental settings. For example, miRNA-21 and miRNA-143, both having well documented roles in cardiac fibroblasts proliferation (67), were predicted to target Arhgap24 and supervillin (Svil). Arhgap24, regulates cell cycle progression and proliferation in lung and renal cancer (68), whereas Svil promotes cell proliferation and survival through regulation of cytokinesis and amplification of stimulus-mediated signaling in various cells

(69). Semaphorin3A (Sema3a) and myosin V (Myo5a) are significantly differentially expressed in recombined versus non-recombined cells and are computationally predicted to be regulated by the RBC-EV cargo RNA miRNA-30e. Sema3a is a key factor expressed by infiltrating leukocytes, shown to reduce cardiac inflammation, facilitate cardiac wound healing, and improve cardiac function in mice after MI (70), whereas Myo5a controls important processes related to cardiomyocyte conductivity and excitability (71). It is feasible to speculate that the transfer of miRNAs within the RBC-EVs may confer protective effects in the RBC-EV recipient cells during IR. However, in the absence of reagents that can inhibit or enhance uptake or release of EVs in vivo without affecting critical cellular processes, we cannot prove the importance of these effects on disease pathogenesis. Ultimately RBC-EV mediated signaling may only be part of complex signaling involving other cell-types and signaling molecules that on balance define post-ischemic cardiac remodeling. Patient-related systemic conditions (e.g., diabetes) may also alter RBC-EV biogenesis or content, affecting this mode of signaling and factoring in the variability of cardiac remodeling seen in human subjects.

The finding that microglia in the brain are targeted by RBC-EVs after MI is intriguing. Microglia are the resident macrophages of the brain and are the first to respond to infection and injury (72) and microglial activation has been reported in the brains of humans with heart disease (73). Under inflammatory conditions, we have demonstrated that EVs released from HCs can overcome BBB and transfer RNA to neural cells (4). Studies by others have shown that EVs derived from RBCs, which are abundant in the toxic forms of $\alpha$-synuclein protein, can cross the BBB under inflammatory conditions, and localize with brain microglia in patients with Parkinson's disease (74). Whether the increase in microglia targeted by RBC-EVs affect processes like depression or cognitive function that have increasingly been recognized as closely associated with ischemic heart disease merits future investigation.

As a proof of concept study, our data were subjected to several limitations. We were unable to rigorously assess differences in the transcriptome profiling between recombined and non-recombined cardiomyocytes in vivo, as we were underpowered for tests within cell types because of the modest number of animals and nuclei profiled in the snRNA-seq. Recognizing this limitation, we focused on evaluating differences between RBC-EV recipient and non-recipient cells including all cell types in the heart, although it should be noted that our top four hits have clearly defined roles in cardiovascular disease (49, 50, 51, 52). Given the robustness of snRNA-seq, it is possible to increase the number of animals in the current study to provide a better resolution in future studies to answer this critical question. The snRNA-seq does not reflect the total mRNA present in a cell, as it excludes RNA outside of the nucleus. In addition, snRNA-seq is designed to detect poly-adenylated transcripts and therefore does not fully explore the presence of RNA molecules that are processed by alternative mechanisms without 3'poly(A) tails. Second, it remains possible that only a proportion of RBC-EVs contain functional Cre, and that the cells marked by GFP positivity (and therefore Cre-mediated recombination) only constitute a proportion of cells that have taken up RBC-EVs; in this case we may be under-estimating the extent of intercellular communication mediated by this mechanism. Third, we did not

characterize the protein or lipid cargo of RBC-EVs that may affect the transcriptomic changes noted in the EV-recipient cells; previous investigations have proposed roles for each of these molecular entities in EV-mediated intercellular signaling. Finally, although it is intriguing that only certain cell types take up RBC-EVs, there remains a paucity of insight into the EV surface markers and cellular receptors that dictate this specificity.

We should also note that this study was designed to investigate the role of RBC-EVs in mediating intercellular communication with heart cells and providing proof-of-concept that snRNA-seq could be leveraged to determine transcriptomic effects of EV-mediated signaling. The study was not designed to be a therapeutic trial to assess the effects of RBC-EVs and their contents on post-ischemic remodeling; such a trial, although of great interest, would require a thorough understanding of the biodistribution, pharmacokinetics and dose–response of different methods of RBC-EV infusion, as well as appropriate controls (such as dummy EVs). Careful consideration of these factors will undoubtedly lead to further studies assessing this hypothesis.

In summary, our model is the first to demonstrate previously unreported communication between EVs released from RBCs and specific organs, both in health and inflammation. Furthermore, our data demonstrate functional changes in the transcript and phenotype of targeted cells, especially in the context of ischemic HF and post-MI remodeling. We hope that this model will be the basis of future investigations into the functional role of this novel signaling mechanism in altering cellular phenotype and modulating disease pathogenesis.

# Materials and Methods

### Animal care and use

All animal studies were approved by the Massachusetts General Hospital Animal Care and Use Committee and under the guidelines on the use and care of laboratory animals for biomedical research published by National Institutes of Health (No. 85-23, revised 1996).The generation of EpoRCre mice (30) and ROSAmT/mG (mT/mG) (31) have been previously described. Wild-type (WT) C57BL6 mice were received from Jackson Laboratory. Maintenance of 12/12 h light–dark cycle was done for all mice. Mice were fed on normal chow diet. Male/female mice of 8–12 wk were fed ad libitum.

For the transfusion experiments, mice were injected i.p. with a single dose of 700 $\mu$l of solution containing complement generated RBC-EVs. The mice were caged in barrier facility and monitored for 7 d before they were euthanized. Blood was collected and organs (heart, lungs, spleen, kidney, liver and brain) were harvested.

For IR surgery, the left anterior descending artery (LAD) was ligated with 7-0 silk. After 30-min LAD occlusion, the LAD ligature was released, and reperfusion was confirmed visually. After 4 wk of reperfusion, mice were euthanized, and organs were harvested for analyses. For the sham surgery, the animals were also intubated, and underwent sternal skin incision without a thoracotomy to minimize the effect of inflammation noted after thoracotomy.

Echocardiography was conducted on conscious mice using a GE Vivid7 with i13L probe (14 MHZ) as described previously (75). Apart from the 2D guided M-mode images of short axis at the papillary muscle, parasternal long-axis views and short-axis views were also recorded. The average of at least 10 measurements was used for every data point from each mouse.

Cardiomyocyte (CM) isolation from IR and sham mice was performed using a previously published protocol (76). Flow sorting of CM population was performed and the enrichment of the viable CMs through this preparation was tested by qPCR of commonly expressed genes in CMs compared with the other fraction from the preparation. CM isolated from EpoR-Cre/mTmG mice after IR surgery were further sorted based on their fluorescence.

EdU (50 mg/kg, subcutaneously) in EpoR-Cre/mTmG mice, after IR, was injected every alternate day for the first 14 d after reperfusion and the animals were euthanized after 6 wk. Sham-operated EpoR-Cre/mTmG mice (without thoracotomy) served as controls.

### Collection of CFP for isolation of EVs

Mice (8–10 wk old) were anaesthetized using 2.4% isoflurane/97.6% oxygen and placed in a supine position on a heating pad (37°C). Cervical dislocation was then conducted in the mice and blood collected from intracardiac puncture using 27 mm gauge needle. The blood was collected in Microvette Micro tube, 1.3 ml, screw cap, EDTA tubes and centrifuged at 1,000$g$ for 15 min. The supernatant was collected in fresh Eppendorf tubes, leaving the buffy coat and the RBC fraction. The buffy coat was discarded, and the RBC fraction was transferred in separate Eppendorf tubes for generation of complement-mediated RBC-EVs. The supernatant from first spin was centrifuged again at 1,500$g$ for 10 min. Then supernatant was CFP, which was taken in fresh Eppendorf tubes leaving the pellet behind and processed for EV isolation.

All human patient-related investigations were conducted in conformation with the principles outlined in the Declaration of Helsinki and duly approved by the relevant institutional review committee (Massachusetts General Hospital). Written informed consent was obtained from all participants before their enrollment in the study. Human venous blood was drawn from healthy volunteers, patients with Type I MI, or with chronic heart failure with the help of 21G needle under aseptic condition and the blood was collected in EDTA-containing vacutainer tubes. The blood was centrifuged at 1,000$g$ for 15 min. The supernatant was collected in fresh 15 ml plastic tubes, leaving the buffy coat and the RBC fraction. The supernatant from first spin was centrifuged again at 1,500$g$ for 10 min and the supernatant was CFP, which was taken in fresh tubes leaving the pellet behind and processed for EV isolation.

### Complement-mediated generation of RBC-EVs

10 µl of EDTA blood was resuspended in HBSS with calcium and magnesium$^{++}$ (HBSS$^{++}$) buffer (1:2 by volume), and centrifuged at 2,500$g$.The supernatant was discarded, and the resulting pellet was resuspended in HBSS$^{++}$. The process was repeated twice. C5b,6 solution (0.18 $\mu$g/ml final concentration in HBSS$^{++}$) was added to the RBC suspension, vortexed, and put on a slow shaker at room temperature for 15 min. C7 (reconstituted to a final concentration of

0.2 μg/ml in HBSS++) was added and put on a slow shaker at room temperature for an additional 5 min. C8 (to a final concentration of 0.2 μg/ml in HBSS++) and C9 (to 0.45 μg/ml in HBSS++) was simultaneously added to the solution and incubated at 37°C for 30 min. After centrifuging the RBC suspension at 2,500g for 10 min, the EV-containing supernatant was collected in a new tube.

## Isolation of EVs from CFP

Isolation of EVs from CFP (mice and human) was performed using a well-described protocol with modifications (77). Three mLs of CFP was resuspended in 5 ml of filtered PBS and underlaid with 2 ml of 60% iodixanol and centrifuged at 100,000g for 2 h. The top 7 ml of supernatant was discarded and the 40% iodixanol with EVs was underlaid over a column gradient consisting of 20%, 10% and 5% iodixanol (suspended in sucrose) and centrifuged at 100,000g for 18 h. Among the 12 fractions, fraction 6–8 was primarily used for our experiments, as fractions 9–10 can also contain lipoproteins. For the functional experiments with EVs (treating HEK293 cells) an extra step of overnight dialysis of the EVs suspended in iodixanol was performed with Spectra-Por Float-A-Lyzer and finally resuspended in filtered PBS.

## Nano-flow cytometry

The principles and detailed workflow of nano-flow cytometry for characterization of EVs was done as previously described in our published work (33). In addition, HBSS++ was used to dilute the plasma EVs to determine the existence of any additional background. Instrument noise, events contributed by the background of HBSS++ used for dilution of samples, and samples were gated appropriately, and gates remained unchanged throughout the experiment. At this setting, EVs were processed at 10,000–20,000 events per second. Antibodies (Abs) used: Glycophorin A (GPA, CD235a [Cat. no. MAB1228-100; R&D Systems]), nonimmune IgG1 (BD Biosciences); secondary antibodies: Alexa Fluor 488 goat anti-mouse IgG, Alexa Fluor 489 goat anti-rabbit IgG (Thermo Fisher Scientific).

## Primary dermal fibroblast and mTmG HEK293 cell lines

Isolation and culture of primary dermal fibroblast was carried out using a previously described protocol (35). On attainment of confluence, the cells were trypsinized and passaged in six well tissue culture dishes (1:5) and kept in tissue culture incubator for 24 h. Subsequently, the dermal fibroblasts were treated with 150 μl of plasma EVs per well. The cells were put in the incubator for 72 h, after which live cell imaging with fluorescence microscopy was performed on them.

Cryopreserved HEK 293T cells were seeded in a Fibroblast growth and culture media (FGCM) (5% FBS, 1% PS, pH 7.4; DMEM). On attainment of 70–80% confluency, the cells trypsinized and split (1:10) into six-well cell culture plates and cultured in FCGM. Amplification of pCA-mTmG and pTriEx-1plasmids was conducted for their use in transfection. The adjusted volume of DNA from pCA-mTmG plasmid were separately solubilized in Lipofectamine 3000 and treated in separate well of six well dishes (2,500 ng DNA per well, 5 μl of P3000

reagent per well, 3.75 μl of Lipofectamine 3000 reagent per well, dissolved in 250 μl of Opti-MEM I Reduced Serum Medium per well). After incubation in FCGM media for 12 h, the antibiotic in media was changed from PS to Blasticidin for antibiotic selection of HEK293 cells. The cells were periodically checked under fluorescence microscope to monitor the fluorescence and confluency of cells, till a stable mTmG HEK293 cell line was generated for future use. The mTmG HEK293 cells were also treated with 100 μl of plasma EVs from EpoR-Cre mice per well and mTmG HEK293 cells treated with DNA from pTriEx-1plasmids (Lipofectamine 3000) served as positive control.

## FACS analysis and sorting

Adult mouse CMs were isolated from sham or IR operated mice's hearts using a Collagenase B (11088807001; Roche), Collagenase D (11088858001; Roche) and Protease XIV (P5147; Sigma-Aldrich) based digestion, following the isolation steps described by Ackers-Johnson et al (2016) Circ Res 119: 909–920. For live-cell sorting and quantitative analysis, adult mouse CMs and HEK293 cells were stained with DAPI (422801; BioLegend) for 30 min in the dark, sorted using a BD FACSAria and analyzed by FlowJo.

## Single nucleus RNA-sequencing

### Experimental sample processing

Whole hearts were harvested from six adult EpoR-Cre/mTmG mice (8–10 wk) after cervical dislocation. The hearts were snap-frozen in liquid nitrogen and stored at –70°C before use. Excised hearts were mounted on a pedestal using a small amount of OCT. Both lower chambers were sectioned in their entirety at a thickness of 70 μm, then lysed in nuclear lysis buffer (250 mM Sucrose, 25 mM KCl, 0.05% IGEPAL-630, 3 mM MgCl$_2$, 1 μM DTT, and 10 mM Tris, pH 8.0). A 1-min centrifugation step at 20g was performed to pellet large debris, followed by filtration through 100 and 15 μm filters (pluriSelect Life Science). Nuclei were rinsed twice in nuclear wash buffer, then resuspended in nuclear resuspension buffer (PBS + 3 mM MgCl$_2$ + 0.01% BSA) at an approximate concentration of 1,000 nuclei per uL. Nuclei were loaded in the 10X Genomics Single Cell 3′ solution (v2 and v3) at an anticipated recovery of 5000 nuclei per device. Library preparation was performed as suggested by the manufacturer. Sequencing was performed by the Genomics Platform of the Broad Institute on an Illumina Nextseq550.

### Single-nucleus data processing

The 10x Genomics toolkit CellRanger was applied to generate count matrices for each sample using standard inputs with the addition of a read trimming step to remove homopolymer repeats and the template switch oligo from demultiplexed reads using the tool cutadapt (78). Reads were aligned to a GRCh38 human pre-mRNA reference to account for the fact that single nucleus RNA-seq was performed. To determine which droplets contained nuclei, we used the *remove-background* tool from CellBender v0.1 with parameters –expected-cells 5000, –z-dim 100, –z-layers 500, –epochs 250, and –total-droplets-included 25,000. CellRanger counts were retained for downstream analysis. Using the tool *scR-Invex* (https://github.com/broadinstitute/scrinvex), we assigned

reads for each sample to exons, introns, or spanning both and calculated the proportion of reads exclusively mapping to exons for each nucleus. Finally, entropy for each nucleus was calculated using the *ndd* python library (https://pypi.org/project/ndd/1.6.3/).

Each of the six samples underwent a nucleus quality control procedure separately to account for distributional differences between samples. In brief, the following four-step procedure was used:

(1) Nuclei were clustered using the standard scanpy (79) clustering procedure with the Louvain algorithm at high resolution (resolution = 2.0).
(2) The distribution of the median values of three quality control metrics were inspected across clusters identified in step 1. This included the fraction of mitochondrial reads, the proportion of reads exclusively mapping to exons, and entropy. Any cluster above the third quartile plus 1.5 times the interquartile range for the first two metrics were removed, and any cluster below the first quartile minus 1.5 times the interquartile range for entropy was removed.
(3) Hierarchical clustering was performed on cluster centroids, and similar clusters were merged together.
(4) Within each merged cluster, low-quality nuclei were identified using an *EllipticEnvelope* (contamination = 0.05) outlier detection algorithm from *scikit-learn* (https://github.com/broadinstitute/scrinvex) based on the fraction of mitochondrial reads, the proportion of reads mapping to exons, and entropy. Detected outliers were removed. Nuclei with log (number of genes) * entropy greater than the 90th percentile across nuclei within the cluster were also removed. Finally, any nuclei with less than 150 genes detected or a fraction of mitochondrial reads greater than 0.05 were excluded.

The above steps were carried out to ensure that variation in quality control metrics between samples and between clusters within samples did not unduly influence which nuclei were removed. Remaining nuclei from each sample were aggregated together and 2,000 highly variable genes were identified across all samples using the *FindVariableFeatures* function in Seurat v3.1.0. To account for technical and biological heterogeneity between samples, the *scVI* algorithm was applied on highly variable genes to compute a latent expression space of 50 latent variables, including a batch indicator for sample. A neighborhood graph was constructed in *scanpy* on this latent space using *sc.pp.neighbors* with n_neighbors = 15 and metric = "cosine," and Leiden clustering was applied with resolution of 0.85 to identify an initial set of clusters. One cluster appeared to have high expression of genes specific to several other clusters raising the concern of possible doublets. To address this, the *Scrublet* algorithm was run for each sample with default parameters. One cluster was identified to have an enriched predicted doublet score and was removed from downstream analysis. After removal of this cluster, a neighborhood graph was reconstructed followed by Leiden clustering and Uniform Manifold Approximation and Projection (UMAP) (https://arxiv.org/abs/1802.03426) with min_dist = 0.1 for visualization. The final map consisted of 42,497 nuclei.

### Marker gene identification
Marker genes were defined for each cluster based on two metrics. First, Area Under the Receiver Operating Characteristic Curve (AUC) was calculated for each gene by treating normalized expression as a prediction and classifying each nuclei as part of the cluster of interest or not. Second, a log(fold-change) comparing normalized expression in the cluster of interest to all other nuclei was computed. Genes with an AUC greater than 0.7 and log (fold-change) greater than one for a given cluster were considered marker genes. These genes were used to help assign cell type labels to each cluster.

### Differential proportions of EGFP versus tomato
To test whether the proportion of EGFP versus Tomato nuclei varied by cell type, a logistic regression model was fit on the three IR samples prepared using SC3Pv3 chemistry. The analysis was restricted to nuclei that exclusively express either EGFP or Tomato and included sample as a fixed effect to control for technical differences between experiments. For alignment, we generated a custom reference "chromosome" which corresponded to the eGFP and tdTomato cassettes contained within the mTmG mice. For all cell types that had at least one EGFP positive and one Tomato positive nucleus in each of the three samples, a contrast was fit comparing the proportion of EGFP positive nuclei in a given cluster to the grand mean using effect coding. In total, 11 cell types were tested, and a Bonferroni significance threshold was set to 0.05/11 = 0.0045. All analyses were carried out in R 3.5.0.

### Differential expression EGFP versus tomato
Differential expression testing was performed between nuclei expressing EGFP and nuclei expressing Tomato regardless of cell type. This analysis was restricted to only the three IR samples prepared using SC3Pv3 chemistry and only nuclei that express exclusively EGFP or Tomato. To account for the correlation in expression from cells in the same sample, a negative binomial or Poisson generalized linear mixed model was used to test for differential expression, including sample as a random effect and the log of total count per nucleus as an offset, similar to Tucker et al (26). Only genes expressed in at least 1% of nuclei from either the EGFP positive or Tomato positive group were considered, leaving 5,045 genes for testing. A Bonferroni significance threshold of $0.05/5,045 = 9.91 \times 10^{-06}$ was used. Generalized linear mixed models were fit using the R package *lme*4.

### RNA isolation, pre-amplification and quantitative real-time PCR (qRT-PCR)

Total RNA was extracted from EV samples and FACS sorted HEK293 cells using mirVana Paris RNA and Native Protein Purification Kit with modifications suggested in an established protocol (80) and further concentrated and purified using RNA Clean and Concentrator-5 with DNase I Set. The RNA was then measured using 4200 TapeStation Instrument using High Sensitivity RNA Screen Tape. For mRNA qPCR cDNA was created from equivalent RNA from each sample with High Capacity cDNA reverse Transcription Kit in Bio-Rad CFX384 qPCR System. For detection of *Cre*-recombinase mRNA, amplification and detection of specific products were carried out using the ExiLENT

SYBR Green master mix in QuantStudio 6 Flex Real-Time PCR System till 45 cycles. Any CT equal or above 40 was considered unreliable and the mRNA was considered not be absent in that sample. For the FACS-sorted HEK293 cells, after the creation of cDNA, the cDNA was then pre-amplified using Taqman PreAmp Master Mix for 20 cycles using the manufacturer's protocol. Further amplification and detection of specific products were performed with Taqman primers and TaqMan Gene Expression Master Mix in QuantStudio 6 Flex Real-Time PCR System till 45 cycles. Any CT equal or above 40 was considered unreliable and the mRNA was considered not be absent in that sample. For HEK293 cells, GAPDH was considered as internal control.

For microRNA, cDNA from total RNA, after clean-up and concentration, was constructed using Universal cDNA Synthesis Kit II in Bio-Rad CFX384 qPCR System (maximum possible RNA input in 20 $\mu$l reaction). Amplification and detection of specific products was carried out using the ExiLENT SYBR Green master mix along with Exiqon miR LNA PCR primer set UniRT, specific to each miRNA, in QuantStudio 6 Flex Real-Time PCR System till 45 cycles.

### ddPCR

In each ddPCR reaction, 4 $\mu$l of cDNAs (diluted 1:10), 10 $\mu$l of QX200 EvaGreen ddPCR Supermix (Bio-Rad), 1 $\mu$l of human Hemoglobin A primers (HbA, Integrated DNA Technologies, Inc.), and nuclease-free water was added in a 20 $\mu$l reaction mixture. A no-template control, where nuclease-free water was added instead of cDNA samples, was also included in the assay. Each ddPCR assay mixture was loaded into an eight-channel droplet generation cartridge (Bio-Rad). Then, 70 $\mu$l of QX200 Droplet generation oil (Bio-Rad) was loaded into the appropriate wells and the cartridge was placed inside the QX200 droplet generator (Bio-Rad). The resulting droplets were carefully transferred to a 96-well plate (Bio-Rad), the plate was then heat-sealed with foil and placed in a conventional thermal cycler (Eppendorf). Thermal cycling conditions were as follows: 95°C for 5 min, then 40 cycles of 95°C for 30 s, and 56°C for 1 min, and three final steps at 4°C for 5 min, 90°C for 5 min, and a 4°C indefinite hold. A ramp rate of 2°C/s was used in all the conditions. At the end of the PCR reaction, droplets were read in the QX200 droplet reader and analyzed using the Quantasoft version 1.7 software (Bio-Rad).

### Pathway analysis

mRNAs found to be significantly differentially expressed in the snRNA-seq dataset (p-adjusted < 0.05) were subjected to over-enrichment analysis against the GO and KEGG databases using the R packages *clusterProfiler* and *pathview*. The background/null set was the full list of genes detected by snRNA-seq and differentially expressed genes with positive fold-changes were considered separately to genes with negative fold-changes.

We also ranked the full list of genes detected by snRNA-seq based on their $\log_2$-fold-change as calculated by DESeq2. This ranked list was then interrogated by gene-set enrichment analysis against GO and KEGG gene-sets.

For both the over-enrichment analysis and gene-set enrichment analysis, adjusted *P*-values < 0.05 were used to filter significantly enriched categories.

### Western blot, immunochemistry, and immunofluorescence staining

Isolated EVs from EpoR-Cre and mTmG mice with ExoQuick Plasma prep and Exosome precipitation kit were resuspended in 30 $\mu$l of filtered PBS. Standard Western blot technique and analysis were carried out on equal amount of protein using Anti-Cre Recombinase (Cat. no.: ab24607, Dilution 1:200; Abcam), CD9 (Cat. no. : 124802, Dilution 1:200; BioLegend) and Ter 119 (Cat. no. : 14592181, Dilution 1:200; Thermo Fisher Scientific) antibodies. The EVs from human plasma isolated through C-DGUC (individual fraction of 1 ml) was again ultracentrifuged for 2 h for precipitation of EVs and subsequently resuspended in 30 $\mu$l of filtered PBS before Standard Western blot technique and analysis were carried out on equal amount of protein using CD9 (Cat. no. : 312102, Dilution 1:1,000; BioLegend) and Anti-Alix (Cat. no. : 634501, Dilution 1:1,000; BioLegend) antibodies.

Harvested organs (heart, lung, liver, spleen, kidney and brain) were were fixed in Paraformaldehyde (4%), overnight. The organs were then cryoprotected with 20% sucrose solution and washed twice with PBS before mounting in OCT and cryo-sectioning into 10-$\mu$m tissue sections. The slides were mounted in DAPI medium before confocal microscopy. For the brain, sagittal sections were cut on a vibratome and kept in PBS at 4°C.

For immunohistochemistry, corresponding sections were treated with acetone/methanol solution before blocking with 3% (wt/vol) BSA in PBS and then incubated with primary antibodies: PHH3 (Cat. no. : 3377S, 1:250; Cell Signaling Technology) and Iba1 (Cat. no. 01919741,1:1,000; Fujifilm Wako Pure Chemical Corporation) and for Anti-NeuN antibody (Cat. no. :01919741; Fujifilm Wako Pure Chemical Corporation) overnight at 4°C. Permeabilization of membranes was done in PBS/Triton X before application of secondary antibody (1: 1,000) for 1 h, and mounting with DAPI for confocal microscopy. For EdU staining, Molecular Probes EdU (5 ethynyl 2 deoxyuridine, Cat. no. A10044; Thermo Fisher Scientific) and Click-iT Plus EdU Alexa Fluor 647 Imaging Kit (Cat. no. C10640; Thermo Fisher Scientific) were used as per manufacturer's protocol. DNA staining was carried out with Hoechst 33342(Cat. no. H3570).

### Isolation of CD235a positive EVs from patient plasma

Mouse monoclonal antibody detecting human Glycophorin A (GPA, CD235a) 100 $\mu$g (Cat. no. MAB1228-100; R&D Systems) was coupled with 6 mg of dynabeads M-270 Epoxy and incubated overnight at 4°C on a roller according to the Dynabeads Antibody Coupling Kit instructions (Cat. no. 14311D; Thermo Fisher Scientific). After washing twice, the antibody-coupled beads were resuspended in 300 $\mu$l of SB buffer along with RNAseOUT, DTT, and PIC solution and then were added to 1 ml of patient plasma followed by rotation at 4°C for 2 h. Magnet induced retrieval of CD235a bound EVs from the dynabeads was performed and the supernatant was discarded.

### Statistical analysis

Data are expressed as the mean ± SEM except for where specified. Normality was tested for all data sets using Shapiro–Wilk test. Statistical analyses were carried out by student unpaired *t* test

(for data that passed normality test) or Mann–Whitney for non-parametric distributions. Differences were considered statistically significant at $P \leq 0.05$ and are indicated with an (*). Analysis and graphs were made using GraphPad Prism V8.

## Data Availability

Single nuclear RNA sequencing data (read counts) are provided in Supplemental Tables (see Tables S1 and S3) and available on SRA (Accession: BioProject PRJNA763810). Small RNA seq data from EVs are provided in Table S5 and available at Dryad with the link: https://doi.org/10.5061/dryad.vhhmgqnv9.

## Supplementary Information

## Acknowledgements

S Das was supported by grants from National Institute of Health (NIH) (NCATS, UG3 TR002878, UG3 HL147353, RO1 HL150401, and R35HL150807); S Momma Deutsche Forschungsgemeinschaft (DFG) (MO2211-2) and Edinger Foundation.

### Author Contributions

N Valkov: formal analysis, investigation, and writing—original draft, review, and editing.
A Das: formal analysis, investigation, and writing—original draft, review, and editing.
NR Tucker: formal analysis, investigation, and writing—original draft, review, and editing.
G Li: investigation.
AM Salvador: formal analysis, investigation, and writing—review and editing.
MD Chaffin: software and formal analysis.
G Pereira de Oliveira Junior: investigation.
I Kur: investigation.
P Gokulnath: investigation.
O Ziegler: investigation.
A Yeri: formal analysis.
S Lu: investigation.
A Khamesra: investigation.
C Xiao: investigation.
R Rodosthenous: methodology.
S Srinivasan: investigation.
V Toxavidis: formal analysis.
J Tigges: formal analysis and methodology.
LC Laurent: formal analysis and writing—review and editing.
S Momma: supervision, investigation, and writing—review and editing.
R Kitchen: formal analysis.
P Ellinor: conceptualization, software, formal analysis, supervision, funding acquisition, and writing—review and editing.
I Ghiran: conceptualization, formal analysis, supervision, methodology, and writing—review and editing.
S Das: conceptualization, formal analysis, supervision, funding acquisition, and writing—original draft, review, and editing.

### Conflict of Interest Statement

The authors declare that they have no competing conflict of interest. S Das is a founding member of Switch Therapeutics that played no role in this study.

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
