## [Reviewer comments · Life Science Alliance]

Life Science Alliance

SnRNA Sequencing Defines Signaling by RBC-Derived Extracellular Vesicles in the Murine Heart

Nedyalka Valkov, Avash Das, Nathan Tucker, Guoping Li, Ane Salvador, Mark Chaffin, Getulio Pereira de Oliveira Junior, Ivan Kur, Priyanka Gokulnath, Olivia Ziegler, Ashish Yeri, Shulin Lu, Aushee Khamesra, Chunyang Xiao, Rodosthenis Rodosthenous, Srimeenakshi Srinivasan, Vasilis Toxavidis, John Tigges, Louise Laurent, Stefan Momma, Robert Kitchen, Patrick Ellinor, Ionita Ghiran, and Saumya Das

DOI: <https://doi.org/10.26508/lsa.202101048>

Corresponding author(s): Saumya Das, Massachusetts General Hospital and Ionita Ghiran, Beth Israel Deaconess Medical Center

Review Timeline:

Submission Date:	2021-02-05
Editorial Decision:	2021-03-27
Revision Received:	2021-08-01
Editorial Decision:	2021-08-17
Revision Received:	2021-09-30
Accepted:	2021-09-30

Transaction Report:

March 27, 2021

Re: Life Science Alliance manuscript #LSA-2021-01048-T

Prof. Saumya Das
Cardiovascular Division of the Massachusetts General Hospital and Harvard Medical School

Dear Dr. Das,

Thank you for submitting your manuscript entitled "Single Nuclear RNA Sequencing Defines Signaling by RBC-Derived Extracellular Vesicles in the Heart" to Life Science Alliance. The manuscript was assessed by expert reviewers, whose comments are appended to this letter. We would like to invite you to submit a revised version of the manuscript that addresses all of the reviewers' points.

We apologize for this extended and unusual delay in getting back to you. As you will see from the reviewers' comments below, the reviewers are interested in these findings, but have also raised a number of significant concerns and questions, all of which must be addressed prior to further consideration of this manuscript at LSA.

Thank you for this interesting contribution to Life Science Alliance. We are looking forward to receiving your revised manuscript.

Sincerely,

Shachi Bhatt, Ph.D.
Executive Editor

- A letter addressing the reviewers' comments point by point.
- An editable version of the final text (.DOC or .DOCX) is needed for copyediting (no PDFs).
- High-resolution figure, supplementary figure and video files uploaded as individual files: See our detailed guidelines for preparing your production-ready images, <https://www.life-science-alliance.org/authors>
- Summary blurb (enter in submission system): A short text summarizing in a single sentence the study (max. 200 characters including spaces). This text is used in conjunction with the titles of papers, hence should be informative and complementary to the title and running title. It should describe the context and significance of the findings for a general readership; it should be written in the present tense and refer to the work in the third person. Author names should not be mentioned.

B. MANUSCRIPT ORGANIZATION AND FORMATTING:

Reviewer #1 (Comments to the Authors (Required)):

In this paper, Valkov and colleagues present an extensive body of data proving the first evidence for intercellular communication by red blood cell EVs (RBC-EVs) as vehicles of functional cargo delivery to cardiomyocytes and other minor cell components in the failing mouse heart. The work is seen as a major advance in the field of EV intercellular communication as it makes use of advanced methods including nano-flow cytometry and snRNA sequencing in gene reporter mice to precisely track cells that are recipients of RBC-EV cargo.

While the long-term functional significance of RBC-EV cargo communication to cardiomyocytes in the failing heart will obviously require additional work, the data presented by Valkov represent to

this reviewer a major achievement in demonstrating cell signaling modulation towards regenerative/repair synthetic pathways that were confirmed independently in the HEK-mTmG reporter cells.

Findings of the study also highlight the capacity for RBC-EVs to target other organs including the spleen and kidney and lungs that have all been previously reported to sequester EVs in prior work by other labs. The lack of RBC-EV targeting any cells within the Liver is remarkable as this organ is recognized as the main sink for EVs injected into mice, and will thus require additional work in future studies that are beyond the scope of this work centered on the heart as a target organ.

Furthermore, while the RBC-EV-signaling data seems to converge on beneficial outcomes in cardiac cell reprogramming, pathological signaling aspects by RBC-EVs should be considered in the Discussion. This could in part help explain adverse outcomes in heart failure among diabetic individual where RBC are exposed to high glucose levels that may confer adverse cell signaling effects.

Collectively, the data convincingly show that RBC-EV can contribute functional cargo delivery to the failing mouse heart to engage transcriptional changes and thus sets the stage for numerous new avenues of research to explore in more detail the beneficial and nefarious consequences of RBC-EV communication in cardiac failure.

Specific comments:

The investigators may consider providing some additional data related to the RBC-EV. This could include an examination of their size and numbers such as by NTA analysis that could thereby provide a more complete presentation of the actual material include the number of RBC-EVs injected into the recipient mTmG recipient mice.

Also, while the Authors provide some discussion as to the presence of double positive "yellow" cells in the mTmG mice injected with the EpoR-Cre EVs, there is no discussion about the "yellow" RBC-EVs themselves. Can the Authors comments on this population of RBC-EVs shown in the nano-FCS plots (Fig2c) and tabulated in Fig.S1c?

The Methods section (page 25) detailing the approach used to isolate the RBC-EVs from mouse/human CFP seems quite tedious. It could therefore be useful for the Authors to add a primary citation of the method to help the readers gain more experimental details into the protocol. Along that note, the Authors may comment on the use of RBC-EVs from fractions 6-8 in the studies, while fractions 9 and 10 also contain positivity for EV markers (Figs S6B).

Can the qRT-PCR data shown in Figure 1D (for Cre mRNA) and 1E (for miR-451) be revised to include a reference mRNA/miRNA to help normalize the presentation of the data.

The word "transfused" used throughout the text to describe the single intraperitoneal injection of RBC-EVs could be reconsider as "infusion".

Reviewer #2 (Comments to the Authors (Required)):

Valkov and co-workers examine erythrocytes-derived extracellular vesicles during cardiac remodeling following ischemia/reperfusion injury. They use a mouse model of erythrocyte reporter mice to study the fate of RBC-EVs following induction of cardiac ischemia in mice and how this

affects gene expression pattern and cardiac remodeling processes.

Overall, the manuscript is interesting and suggests that erythrocytes or their components participate in cardiovascular remodeling processes and that RBC-EV formation and cargo transmission may work as a new way of intercellular communication. However, I have several methodological questions or concerns and am missing mechanistic insights regarding the generation, uptake and action in the setting of murine cardiac ischemia/reperfusion injury. Also, there is no information on how RBC-EV uptake affects cardiac remodeling processes (no control group without RBC-EV injection).

In addition, I have the following specific comments:

1. Please clarify whether control mice have been crossed with ErCre tg mice and carry the wildtype Cre allele or whether this group represents mice expressing only the floxed gene. If the latter, could the absence of the Cre allele have contributed to the absence of green signals in the target cells (e.g. in Figure 2D)?
2. Having generated this novel reporter mouse line and assuming that no other cells are transduced (except some platelets), why were EV generated non-physiologically ex vivo and then injected i.p.? Why were endogenous RBC-EVs (generated in response to injury) not examined or RBC-EVs injected directly into the heart or the cardiac circulation? As of now, questions remain, for example how many RBC-EVs are reaching the heart after i.p. injection or if other, indirect or systemic effects may have played a role. It also seems that a lot of the EVs are entrapped in the lungs (Suppl Fig. S3).
3. The author show that EVs carry Cre recombinase protein, but never show that Cre mRNA, protein or remnants of erythrocyte EVs (e.g. RBC-specific membrane proteins or other factors) are transmitted to the target cell. This should be examined to confirm the transduction with RBC-EVs and to provide a plausible explanation for the change in target cell alterations.
4. Also regarding the fate of EVs after targeting the recipient cells: Can the presence of EVs in the target cells be traced by using fluorescent lipid membrane dyes in vitro?
5. What does cardiac ischemia affect endogenous RBC-EV generation, what is the stimulus for RBC-EV uptake in different types of tissues of uninjured mice and how is this affected by cardiac ischemia/reperfusion injury?
6. What are the downstream mechanisms: Which components of RBC-EVs are transferred to the target cells and involved in the phenotype and changes in target cell gene expression? Is this a direct or an indirect effect? Are specific factors transferred? This remains unexplained. To the same end, the authors performed snRNA-seq of target cells, but not of the RBC-EVs to investigate their role as source and cause of transcriptional regulation.
7. Only the effects of cardiac injury itself were examined, but not whether this is affected by the RBC-EV injection. There is no control group of mice undergoing surgery without RBC-EV injection. In my opinion, this is the most interesting, relevant question, which is not at all examined.
8. Throughout the manuscript, but particularly in Figures 3B, 3G and 5 (but also in Figure S4E, S7B), there is a large degree of data variation and overlap between the groups together with small number of animals examined (e.g. n=3 in Figure 5A), which is not sufficient for reliable statistical analysis. How solid are the data? Which test was used?
9. Figures S5: mTomato-positive cells can also be observed in uninjured mT/mG mice amounting to 30%. How can this be explained in the absence of Cre? In the same figure, please provide quantitative data for the results shown in Figures S5C.
10. Control groups are also missing in several other panels, e.g. Figure S6D, S7B.

Minor comments:

- The manuscript title should be modified to better reflect the contents of the study and contain a specific statement. Moreover, it should be clearly stated, in the title and the abstract, that this study was performed in mice.
- The total number of citations is quite high (115), please try to reduce.
- There appears to be a problem with the organisation of Figure 4, at least on the pdf for review. Please check.

Point by point response to reviewer comments:

Reviewer #1 Comments:

In this paper, Valkov and colleagues present an extensive body of data proving the first evidence for intercellular communication by red blood cell EVs (RBC-EVs) as vehicles of functional cargo delivery to cardiomyocytes and other minor cell components in the failing mouse heart. The work is seen as a major advance in the field of EV intercellular communication as it makes use of advanced methods including nano-flow cytometry and snRNA sequencing in gene reporter mice to precisely track cells that are recipients of RBC-EV cargo.

While the long-term functional significance of RBC-EV cargo communication to cardiomyocytes in the failing heart will obviously require additional work, the data presented by Valkov represent to this reviewer a major achievement in demonstrating cell signaling modulation towards regenerative/repair synthetic pathways that were confirmed independently in the HEK-mTmG reporter cells.

We thank the reviewer for their careful reading of the manuscript and enthusiastic support of our findings. We hope that dissemination of this work to the scientific community will allow for use of these technologies by other groups to advance the field of EV signaling further.

Findings of the study also highlight the capacity for RBC-EVs to target other organs including the spleen and kidney and lungs that have all been previously reported to sequester EVs in prior work by other labs. The lack of RBC-EV targeting any cells within the Liver is remarkable as this organ is recognized as the main sink for EVs injected into mice, and will thus require additional work in future studies that are beyond the scope of this work centered on the heart as a target organ.

We thank the reviewer for this interesting observation. In the liver, the Kupfer cells appear to be a major target of EVs when assessed by other methods (Blood. 2005 Mar 1;105(5):2141-5). It is likely that in these tissue resident macrophage cells, RBC-EVs are captured and targeted to the lysosomal system where cre is digested and therefore cannot affect recombination in the nucleus. Based on the reviewer's comment, this is now noted in the manuscript, page 18 (lines 16-19). Nonetheless, this needs to be explored further, but is not in the scope of this manuscript.

Furthermore, while the RBC-EV-signaling data seems to converge on beneficial outcomes in cardiac cell reprogramming, pathological signaling aspects by RBC-EVs should be considered in the Discussion. This could in part help explain adverse outcomes in heart failure among diabetic individual where RBC are exposed to high glucose levels that may confer adverse cell signaling effects.

This is another excellent observation. We now note in the discussion (see page 22, line 5-13), that patient characteristics (such as diabetes or anemia of chronic disease) may affect the quantity, cargo, targeting and function of RBC-EVs.

Collectively, the data convincingly show that RBC-EV can contribute functional cargo

delivery to the failing mouse heart to engage transcriptional changes and thus sets the stage for numerous new avenues of research to explore in more detail the beneficial and nefarious consequences of RBC-EV communication in cardiac failure.

We agree that this study can serve as a platform for further investigations into the different cargoes of RBC-EVs and how they may affect the function of the different cardiac resident cells (both deleterious and beneficial).

Specific comments:

The investigators may consider providing some additional data related to the RBC-EV. This could include an examination of their size and numbers such as by NTA analysis that could thereby provide a more complete presentation of the actual material include the number of RBC-EVs injected into the recipient mTmG recipient mice.

We have provided data on RBCs generated from WT or RBC expressing cre (see Fig S1). The presence of cre does not affect the size or number of EVs generated by complement activation (the methodology used in the manuscript). Based on these numbers, we injected approximately 1.23×10^9 EV particles predominantly in the 100-150 nm range (see Fig S1C). Furthermore, electron microscopy confirmed the morphology of the RBC-EVs generated (Fig S1B.)

Also, while the Authors provide some discussion as to the presence of double positive "yellow" cells in the mTmG mice injected with the EpoR-Cre EVs, there is no discussion about the "yellow" RBC-EVs themselves. Can the Authors comments on this population of RBC-EVs shown in the nano-FCS plots (Fig2c) and tabulated in Fig.S1c?

We apologize for the lack of clarity about the 'yellow' RBC-EVs in the manuscript. The cre-mediated recombination of mT/mG in RBC is a temporal function and thus the fluorescence of a given EV will depend on the half-life of the mTomato in these cells; the yellow or double-positive RBC-EVs (mGFP positive/mTom positive) is due the presence of both proteins in the cell membrane. Finally, as we used homozygote mTmG mice, recombination in one allele and not the other, may lead to double positive cells, as well as double positive EVs. Both of these possibilities are discussed (page 7, line 1-3). This is also seen in the single nuclear RNAseq, where there are cells with both mTom and mGFP transcripts in the EV recipient cells (page 18, lines 22-page 19, line 2) .

The Methods section (page 25) detailing the approach used to isolate the RBC-EVs from mouse/human CFP seems quite tedious. It could therefore be useful for the Authors to add a primary citation of the method to help the readers gain more experimental details into the protocol. Along that note, the Authors may comment on the use of RBC-EVs from fractions 6-8 in the studies, while fractions 9 and 10 also contain positivity for EV markers (Figs S6B).

We wish to thank the reviewer for the suggestion. We have added the citation in that designated area of the manuscript for readers to refer if they are interested in a detailed protocol (see page 27, line 2). Fractions 6-8 were used for experiments as these fractions contain pure EVs and fractions 9 and 10 may have other components such as lipoproteins. Thus, while fractions 9/10

may also contain EVs, we chose not to use them (this is explained on page 27, line 7).

Can the qRT-PCR data shown in Figure 1D (for Cre mRNA) and 1E (for miR-451) be revised to include a reference mRNA/miRNA to help normalize the presentation of the data.

This is an excellent point. The main issue is that there is no universal 'normalizer' for EV-RNAs that can be agreed upon. Therefore, we have conducted droplet digital PCR from RBC-EVs of wild-type or mice with cre expression in RBCs (this is shown in Supplemental Fig 1G). These data show that miR-451 is abundant in EVs from both mice, but no cre mRNA is detected in either.

The word "transfused" used throughout the text to describe the single intraperitoneal injection of RBC-EVs could be reconsidered as "infusion".

Thank you for the suggestion. We have changed the word to infusion for the single intraperitoneal injections.

Reviewer #2 (Comments to the Authors (Required)):

Valkov and co-workers examine erythrocytes-derived extracellular vesicles during cardiac remodeling following ischemia/reperfusion injury. They use a mouse model of erythrocyte reporter mice to study the fate of RBC-EVs following induction of cardiac ischemia in mice and how this affects gene expression pattern and cardiac remodeling processes.

Overall, the manuscript is interesting and suggests that erythrocytes or their components participate in cardiovascular remodeling processes and that RBC-EV formation and cargo transmission may work as a new way of intercellular communication. However, I have several methodological questions or concerns and am missing mechanistic insights regarding the generation, uptake and action in the setting of murine cardiac ischemia/reperfusion injury. Also, there is no information on how RBC-EV uptake affects cardiac remodeling processes (no control group without RBC-EV injection).

We thank the reviewer for his comments. We would stress that this initial study was to demonstrate the feasibility of this model to study EV-mediated intercellular communication. Most notably this is the first study that has used single nuclear RNAseq to profile cells that were targeted by EVs, thereby providing the platform to study the mechanistic questions raised by the reviewer (which is beyond the scope of this manuscript). We do show using both fluorescent mapping and snRNAseq that different cell types are targeted by RBC-EVs, and targeted cells have alteration in gene expression that may be consistent with the observed phenotype. Although of great interest to us, this study was not designed to be a therapeutic trial (i.e. to determine if RBC-EVs may be beneficial for remodeling); in this case, control animals with dummy EVs or no EVs would be the appropriate controls (as a sham group). As we were only comparing recombined (targeted) to non-recombined (not targeted) cells in the same heart

using *sncRNAseq*, these controls were not done. This is noted in the limitations section (page 23: lines 16-24).

In addition, I have the following specific comments:

1. Please clarify whether control mice have been crossed with ErCre tg mice and carry the wildtype Cre allele or whether this group represents mice expressing only the floxed gene. If the latter, could the absence of the Cre allele have contributed to the absence of green signals in the target cells (e.g. in Figure 2D)?

We apologize for any confusion. The EpoR-Cre mouse is a purely transgenic mouse where cre is expressed under the EpoR promoter to restrict expression to erythrocyte progenitor cells. There is no lox site in these mice. As cre is not a mouse gene (it is an exogenous yeast gene), there is no 'wild-type cre' allele. Mice homozygous for EpoRCre were lethal in utero, hence the experimental group of EpoRCre was Cre positive for one allele (heterozygous). The presence of this single cre allele leads to expression of cre protein in both cells and EVs. When mated with the mTmG mice (where the mT gene is floxed), the expression or transfer of functional cre will lead to recombination and excision of the mTomato signal. This is shown in the schematic in Fig 1A. We do agree with the reviewer that the efficiency of cre-mediated may be variable (as noted, especially in vitro, the efficiency of EV-mediated transfer of cre is poor (see pages 7: lines, 14, lines 9-10,)).

2. Having generated this novel reporter mouse line and assuming that no other cells are transduced (except some platelets), why were EV generated non-physiologically ex vivo and then injected i.p.? Why were endogenous RBC-EVs (generated in response to injury) not examined or RBC-EVs injected directly into the heart or the cardiac circulation? As of now, questions remain, for example how many RBC-EVs are reaching the heart after i.p. injection or if other, indirect or systemic effects may have played a role. It also seems that a lot of the EVs are entrapped in the lungs (Suppl Fig. S3).

We appreciate the opportunity to clarify. In fact, the double transgenic (EpoR-cre/mTmG mice) model measures the effects of endogenously generated RBC-EVs either at baseline (see Fig 1B,C) or following ischemia-reperfusion (Fig 3C,D). In these model, RBC-EVs are generated (likely by complement activation) in vivo and target various cells as we measured resulting in transfer of functional cre as well as likely other cargoes leading to changes in the transcriptome (Fig 4). The EV infusion experiments are conducted to demonstrate that functional transfer of cre from RBC-EVs is sufficient to mediate recombination in the target cells (and it is not from ectopic expression of cre or cell fusion as are theoretical possibilities in the double transgenic mice). We have clarified this in the text. Finally, ex-vivo generation of the EVs from the EpoR-cre RBCs allows for generation of a large number of EVs that can be quantified and infused (see new supplemental Fig 1E,F). The points about biodistribution of infused EVs is important in the context of a therapeutic trial (which our study was not), but is noted (see response to point 7).

3. The author show that EVs carry Cre recombinase protein, but never show that Cre mRNA, protein or remnants of erythrocyte EVs (e.g. RBC-specific membrane proteins or

other factors) are transmitted to the target cell. This should be examined to confirm the transduction with RBC-EVs and to provide a plausible explanation for the change in target cell alterations.

An important consideration that the reviewer has brought to notice here. We aim to explain the findings from the data already provided in the manuscript. From Figure 1, we show that RBC-EVs carry cre protein and not Cre-mRNA, although it is possible that some of the effects may be from fusion or uptake of RBC remnants by target cells. To rule out the effects of remnants of RBC and cell fusion, we performed infusion experiments where RBC-EVs from EpoR-Cre were infused in mT/mG mouse to verify that it is the cre protein in these EVs that are taken up by target cells and mediate the recombination (mGFP expression). Note that the pattern is similar (not statistically different). However, we do concede this other mode of cre recombination in the limitations section (page 18, lines 8-11).

4. Also regarding the fate of EVs after targeting the recipient cells: Can the presence of EVs in the target cells be traced by using fluorescent lipid membrane dyes in vitro?

Some studies have used membrane dyes to label EVs and demonstrate their uptake. However, aggregation of lipophilic dyes remains a problem and a confounder, (see PMID 32533028, As we were interested in the functional aspects of EV transfer, we did not study using this methodology. Furthermore, this method cannot be used in vivo (the focus of our study).

5. What does cardiac ischemia affect endogenous RBC-EV generation, what is the stimulus for RBC-EV uptake in different types of tissues of uninjured mice and how is this affected by cardiac ischemia/reperfusion injury?

Cardiac injury can lead to complement activation that is a key generator of RBC-EVs (see PMID 29696020) noted on page 10, lines 3-4. The receptors for EV uptakes have not been identified, so it is not possible to determine how this changes with IR. We do think this is a very interesting question, but cannot be addressed within the scope of this study.

6. What are the downstream mechanisms: Which components of RBC-EVs are transferred to the target cells and involved in the phenotype and changes in target cell gene expression? Is this a direct or an indirect effect? Are specific factors transferred? This remains unexplained. To the same end, the authors performed snRNA-seq of target cells, but not of the RBC-EVs to investigate their role as source and cause of transcriptional regulation.

These are again important questions for the field and would constitute entire studies on their own. Protein, lipid, and RNA cargoes have been shown and hypothesized to effect some of the effects of EV-mediated communication and profiling of cargo is a large study onto itself. Given the very low numbers of murine EVs that can be harvested for downstream analysis (such as RNAseq) and the lack of markers specific for murine RBC-EVs, we instead performed small RNAseq on CD235 pull-down from human plasma: CD235 is specific for RBC-EVs. In the revised manuscript, we show the top RBC-EV small RNAs and computational analysis to show their possible interaction with the pathways of interest (see Figs S9 and S10, Results page 16-

16 and discussion of this on page 21, lines 8-25).

7. Only the effects of cardiac injury itself were examined, but not whether this is affected by the RBC-EV injection. There is no control group of mice undergoing surgery without RBC-EV injection. In my opinion, this is the most interesting, relevant question, which is not at all examined.

This is certainly an interesting therapeutic question: do RBC-EVs affect or modulate cardiac injury? In this case, appropriate controls would certainly be mice with RBC-EV versus dummy EV injections. Whether RBC-EVs can benefit post-ischemic remodeling would be important in the context of a therapeutic trial. However, we have clearly stated that this study was not designed to be a therapeutic trial of RBC-EVs (see page xx), but was designed to answer a different question, i.e. do RBC-EVs communicate with cardiac cells, which types of cardiac cells, and how does this communication affect gene expression. It is well known that mice generally remodel poorly (as we have shown), and is apparent in the double transgenic mice (all of which show a decline in EF). It would be interesting to assess if infusing large number of RBC-EVs (or modifying content) can affect cardiac remodeling, but this would require first, a careful study of biodistribution, dose response and standard PK data, all beyond the scope of this manuscript. However, we have noted this in the discussion (page 23, lines 14-22).

8. Throughout the manuscript, but particularly in Figures 3B, 3G and 5 (but also in Figure S4E, S7B), there is a large degree of data variation and overlap between the groups together with small number of animals examined (e.g. n=3 in Figure 5A), which is not sufficient for reliable statistical analysis. How solid are the data? Which test was used?

While for some of the experiments that were not central to the hypothesis, our sample size was small, for the key experiments, we conducted sufficient number of experimental repeats to support our conclusions (e.g. Fig 3B had 8 animals for IR, 7 for sham, see page 53, line 14; Fig 3G had 8 total animals and is not central to the hypothesis; Fig S7B was removed as it was not adding much to the overall conclusions; Fig S4E was also not central and was shown to demonstrate no significant differences in other organs). While the sample size can always be increased, these were time intensive experiments subjected to standard statistical analysis (testing for normality and the appropriate comparison test as noted in the Figure legends and methods). We may certainly be underpowered to detect smaller differences, and this is acknowledged in the discussion (page 22, lines 15-19).

9. Figures S5: mTomato-positive cells can also be observed in uninjured mT/mG mice amounting to 30%. How can this be explained in the absence of Cre? In the same figure, please provide quantitative data for the results shown in Figures S5C.

We appreciate the opportunity to clarify. The control (uninjured) group in the transfusion experiment were mTmG mice transfused with EpoR-Cre RBC-EVs (therefore containing functional cre protein). Hence it would be expected that there would be some degree of recombination at baseline (as we have shown in Fig 2 and Fig S5 in the double transgenic mice). The majority of CMs would still be mTomato-positive (not recombined), and at baseline a

small percentage, i.e. 3-5% depending on the exact model) receive EVs and undergo recombination (mGFP-positive) as shown here. The efficiency of recombination of infused EVs versus endogenously generated EVs (as in the double transgenic mice) may be different, but this was not a subject of this study. This degree of recombination in cardiomyocytes (i.e. those that take up these cre-containing EVs increase with IR in the double transgenic mice, as shown in Fig 3). The infusion experiments serve as a control for ectopic expression of cre or cell fusion to explain the cre-mediated recombination. These points are now clarified on page 11, lines 13-17). The reviewer is correct in that the percentage of mGFP positive (or recombined) cells appears to be higher with this mode of delivery (iv infusion) compared to endogenous RBC-EVs in the double transgenic mice. Although this may be an important point for future studies that may leverage RBC-EVs as delivery vehicles, it is beyond the scope of this manuscript. The microscopy data was done to complement our nanoFCS and visually demonstrate the presence of mGFP positive cells that are cardiomyocytes and not used for cell counting (this has been clarified).

10. Control groups are also missing in several other panels, e.g. Figure S6D, S7B.

The definition of control group has been defined in each figure legend. For S7D (previously was S6D as above), the mTmG HEK293 reporter cells transfused with EpoR-cre RBC-EVs resulting in a low (5-8%) recombination resulting in mGFP positive cells; the non-recombined cells remain mTomato positive and do not have any mGFP positivity as they were never exposed to cre. Hence these are not shown here (as there is 0% recombination). This is clarified in the Figure Legend for Fig. S7. As we were only assessing fold change of transcripts between recombined vs non-recombined cells, we did not treat these cells with dummy EVs or PBS control (which would not have led to any recombination). On further examination, the previous Figure S7 was not contributing to the results and therefore, it is now omitted in the revision.

Minor comments:

- **The manuscript title should be modified to better reflect the contents of the study and contain a specific statement. Moreover, it should be clearly stated, in the title and the abstract, that this study was performed in mice.**

We have clarified that this was a murine model in the title and abstract (see page 1 and page 2, line 8). We are happy to work with the editor to modify the title.

- **The total number of citations is quite high (115), please try to reduce.**

We have reduced the number of citations by 35.

- **There appears to be a problem with the organisation of Figure 4, at least on the pdf for review. Please check.**

This has been corrected.

August 17, 2021

RE: Life Science Alliance Manuscript #LSA-2021-01048-TR

Prof. Saumya Das
Cardiovascular Division of the Massachusetts General Hospital and Harvard Medical School
Simches 3
185 Cambridge Street
Boston, MA 02114

Dear Dr. Das,

Thank you for submitting your revised manuscript entitled "SnRNA Sequencing Defines Signaling by RBC-Derived Extracellular Vesicles in the Murine Heart". We would be happy to publish your paper in Life Science Alliance pending final revisions necessary to meet our formatting guidelines.

- please add ORCID ID for the corresponding (and secondary corresponding) author-you should have received instructions on how to do so
- please upload your main and supplementary figures as single files
- if possible please provide each figure as one file
- please add your main, supplementary figure, and table legends to the main manuscript text after the references section
- please note that titles in the system and manuscript file must match
- please make sure the author order in your manuscript and our system match
- please consult our manuscript preparation guidelines <https://www.life-science-alliance.org/manuscript-prep> and make sure your manuscript sections are in the correct order
- please use the [10 author names, et al.] format in your references (i.e. limit the author names to the first 10)
- please be sure that you added all Authors in the Author Contribution section in your manuscript text
- please upload your Tables in editable .doc or excel format
- please add callouts for Figure S9A-C to your main manuscript text
- please indicate scale bar size in Legends for Figures S9B
- please add a Data Availability section if the single nucleus RNA-sequencing data has been deposited

FIGURE CHECKS:

- readability of scale bars through all figures needs to be improved
- missing scale bars for Figure 3F

LSA now encourages authors to provide a 30-60 second video where the study is briefly explained. We will use these videos on social media to promote the published paper and the presenting author. Corresponding or first-authors are welcome to submit the video. Please submit only one

video per manuscript. The video can be emailed to contact@life-science-alliance.org

A. FINAL FILES:

B. MANUSCRIPT ORGANIZATION AND FORMATTING:

****Reviews, decision letters, and point-by-point responses associated with peer-review at Life Science Alliance will be published online, alongside the manuscript. If you do want to opt out of**

having the reviewer reports and your point-by-point responses displayed, please let us know immediately.**

Sincerely,

Reviewer #1 (Comments to the Authors (Required)):

Valkov and colleagues present the first evidence for intercellular communication by red blood cell EVs (RBC-EVs) as vehicles of functional cargo delivery to cardiomyocytes and other minor cell components in the failing mouse heart.

The work is a major advance in the field of EV intercellular communication as it makes use of advanced methods including nano-flow cytometry and snRNA sequencing in gene reporter mice to precisely track cells that are recipients of RBC-EV cargo.

The Authors have addressed all of my concerns and I have no further issues to raise as the study is an outstanding achievement in EV biology as it relates to intercellular communication in cardiac failure.

Reviewer #2 (Comments to the Authors (Required)):

The authors could clarify my questions and have addressed my concerns. This is a very interesting, relevant study and I have no further comments.

September 30, 2021

RE: Life Science Alliance Manuscript #LSA-2021-01048-TRR

Prof. Saumya Das
Massachusetts General Hospital
Simches 3
185 Cambridge Street
Boston, MA 02114

Dear Dr. Das,

Thank you for submitting your Research Article entitled "SnRNA Sequencing Defines Signaling by RBC-Derived Extracellular Vesicles in the Murine Heart". It is a pleasure to let you know that your manuscript is now accepted for publication in Life Science Alliance. Congratulations on this interesting work.

DISTRIBUTION OF MATERIALS:

Again, congratulations on a very nice paper. I hope you found the review process to be constructive and are pleased with how the manuscript was handled editorially. We look forward to future exciting submissions from your lab.

Sincerely,
